# Greenhouse warming and internal variability increase extreme and central Pacific El Niño frequency since 1980

Ruyu Gan [1,2,3], Qi Liu [4,5] ✉, Gang Huang [1,2,3] ✉, Kaiming Hu [1,6] & Xichen Li [7]

El Niño has been recorded to change its properties since the 1980s, characterized by more common extreme El Niño and Central Pacific (CP) El Niño events. However, it is still unclear whether such change is externally forced or part of the natural variability. Here, we find that the frequency of the extreme and CP El Niño events also increased during the period 1875–1905, when the anthropogenic $CO_2$ concentration was relatively lower, but with a positive phase of the Atlantic Multidecadal Oscillation (AMO). Models and palaeoclimate proxies reveal that a positive AMO enhances the zonal sea surface temperature gradient in the CP, which strengthens zonal advective feedback, favoring extreme and CP El Niño development. Moreover, we estimate that internal variability contributed to ~65% of the increasingly extreme and CP El Niño events, while anthropogenic forcing has made our globe experience ~1 more extreme and ~2 more CP events over the past four decades.

El Niño, the warm phase of the most dominant interannual climate variability, has changed its behavior since the 1980s[1-4]. Extreme El Niño events have occurred frequently in the past 40 years, at a rate of one event per 13 years, usually causing reorganizations of atmospheric convection and inducing severe climatic disruptions around the globe, such as disastrous floods and droughts across the Pacific region[5-7]. At the same time, the Central Pacific (CP) El Niño[8-10] (also termed the Dateline El Niño[11], El Niño Modoki[12], or warm pool El Niño[13]), which is characterized by the peak ocean warming in the central equatorial Pacific, has become more common[13-16] and the canonical Eastern Pacific (EP) El Niño, which is characterized by the strongest sea surface temperature anomalies (SSTA) located in the far eastern equatorial Pacific, has become less frequent[8-10,12]. The movement of the sea surface temperature (SST) anomaly center shifts atmospheric circulation and convection, causing different climate impacts around the globe[9,11,14,17].

Several studies using climate models have projected that El Niño amplitude increases[7] and CP El Niño variability increases[14] under greenhouse warming and have suspected that the observed El Niño behavior changes could be a consequence of anthropogenic warming. With the buildup of greenhouse gases, the eastern equatorial Pacific warmed faster than the surrounding regions, and the thermocline in the equatorial Pacific flattened, facilitating more occurrences of extreme El Niño and CP El Niño events[7,14]. The enhanced response of atmospheric vapor to air temperature under greenhouse warming also increases the extreme El Niño[18]. Multiple palaeo-El Niño proxies also provide evidence for the impact of the climate change on the observed change in El Niño properties. These include enhanced El Niño variability during the late twentieth century relative to the preindustrial period[15,19-21] and more frequent CP events in recent decades relative to the preindustrial era[15,22]. In addition to anthropogenic forcing, natural

[1]State Key Laboratory of Numerical Modeling for Atmospheric Sciences and Geophysical Fluid Dynamics, Institute of Atmospheric Physics, Chinese Academy of Sciences, Beijing, China. [2]Laboratory for Regional Oceanography and Numerical Modeling, Qingdao National Laboratory for Marine Science and Technology, Qingdao 266237, China. [3]University of Chinese Academy of Sciences, Beijing 100049, China. [4]School of Atmospheric Sciences, Nanjing University, Nanjing 210023, China. [5]Joint International Research Laboratory of Atmospheric and Earth System Sciences, Nanjing University, Nanjing 210023, China. [6]Collaborative Innovation Center on Forecast and Evaluation of Meteorological Disasters (CIC-FEMD), Nanjing University of Information Science & Technology, Nanjing, China. [7]International Center for Climate and Environment Sciences, Institute of Atmospheric Physics, Chinese Academy of Sciences, Beijing, China. ✉e-mail: qiliu@nju.edu.cn; hg@mail.iap.ac.cn

variability also plays a role in the change in El Niño properties[23,24]. Preindustrial model simulations show that El Niño characteristics exhibit strong interdecadal and intercentennial modulation in the absence of external forcing[23]. Palaeo-El Niño reconstructions also show a wide range of El Niño variance over the past 7000 year[21] and find the El Niño variabilities in both the early 1900s and recent decades are relatively higher than preindustrial levels[25], highlighting the role of internal variability. Observations and preindustrial-control model simulations show that Atlantic Multidecadal Oscillation (AMO), a long-lived basin-wide warming or cooling in the Northern Atlantic that generally persists for 60–80 years, could modulate El Niño amplitude[26–28] and El Niño type[29,30].

It is still unclear whether the frequent occurrence of extreme El Niño events and CP El Niño events in the last 40 years is part of natural variability[23,24] or a consequence of global warming[1,7,14]. In order to investigate this, we can analyze past El Niño records to see whether there are other periods with frequent extreme El Niño and CP El Niño occurrences like the past 40 years, but without strong anthropogenic forcing. If this is the case, we can deduce that factors other than anthropogenic activity must play a role in the decadal transition of El Niño types. In this study, we examine multiple long-term instrumental SST datasets and apply cluster analysis to classify El Niño events. We show that there exists another extreme El Niño and CP El Niño epoch, around year 1900, with similar spatial and temporal evolution, dynamic processes, and climate impacts as those that occurred in the last 40 years. The results suggest a role of internal variability. Moreover, we found that both the two periods with increased extreme El Niño and CP El Niño events coincide with the positive phase of the AMO. Then, we investigate the influence of internal variability associated with the AMO on El Niño multidecadal modulation using outputs from the fifth and sixth phases of the Coupled Model Intercomparison Project (CMIP5 and CMIP6) and palaeoclimate proxies. Furthermore, we quantify the contribution of anthropogenic forcing and internal variability to the recently observed El Niño diversity based on a statistical model. Our results highlight that both internal variability associated with the AMO and anthropogenic forcing contribute to the changes in El Niño properties in recent decades.

## Results

### A period beyond the last four decades with frequent extreme and CP events

We applied a cluster analysis to the evolution from the onset to the development of 38 El Niño events for the period 1871 to 2017 using a "merged" SST dataset from HadISST1[31], Extended Reconstructed SST version 5 (ERSSTv5)[32], and Kaplan Extended SST[33] (see Methods). Based on the cluster analysis, El Niño events were classified into 4 physically meaningful clusters: (1) extreme, (2) EP, (3) CP, and (4) successive El Niño events (Fig. 1 and Supplementary Fig. 1). The four types of El Niño in this study are mutually exclusive. This categorization distinguishes extreme from moderate events, thus the "EP" in this study refers to "moderate EP". The four types of identified El Niño events are consistent with ref. [34] during the common period (1901–2017). We focus on the extreme, EP, and CP types of El Niño events in this study. It is intriguing that almost all extreme and CP events occurred in the two periods. One is the period from the 1980s to the present, in which ~8 CP and 3 extreme events exist, and the other is the period 1870s to the 1900s, among which 3 extreme and 3 CP events consecutively occur (Fig. 2a). In contrast, EP events mainly occurred in the 1900s–1930s and 1950s–1980s. The two increased extreme and CP event periods are also detectable from the HadISST1, ERSSTv5, and Kaplan SST datasets (Supplementary Fig. 2a–c). In the three datasets, the numbers of extreme El Niño events are 3, 2, and 2 during the period from the 1870s to the 1900s and 3, 3, and 3 during the period from the 1980s to the present, respectively. For CP El Niño, the number are 3, 3, and 3 during the period from 1870s to the 1900s and 7, 8, and 7 during

the period from the 1980s to the present. The slight difference should arise from data uncertainty.

The extreme and CP El Niño events that occurred in the 1870s to 1900s were similar to those in the 1980s to present in both their spatiotemporal evolution of SSTA and dynamic processes. We compare the spatiotemporal evolution of extreme El Niño and CP El Niño events for the two periods using the "merged" SST dataset. The evolution of the extreme El Niño events in the two periods is similar: the warm SST anomaly starts to develop in the western Pacific (WP) during the preceding boreal winter and then propagates eastward with a rapid basin-wide development in the boreal spring. Later, a warm anomaly occurs in the far eastern Pacific during boreal spring and then propagates westward. The maximum intensity occurs around 120°W in December (Fig. 1a, b; Supplementary Fig. 3a). Strong westerly anomalies develop in the western Pacific in the preceding boreal winter and spring (Supplementary Fig. 4a, b). Coupled with pronounced westerly, convective, and warm SST anomalies near the dateline (180°E) (Supplementary Fig. 4e, f), these anomalies are conducive to initiating an eastward propagation of SST anomalies[35–37]. Moreover, we conducted an ocean mixed-layer heat budget analysis of the mixed-layer ocean temperature averaged over the central-eastern Pacific (5°S–5°N, 180°–80°W) region during the El Niño development phase. It shows that during the onset phase, the zonal advective feedbacks are dominant for the extreme events in these two periods (Fig. 1c), and this result is robust, with a total of 5 of 6 extreme events in these two periods (83.3%) supporting it (Supplementary Table 1).

The CP El Niño events in the two periods also show a similar evolution and feature initial warming in the western Pacific and eastward propagation from the western Pacific from early boreal summer to boreal winter (Fig. 1d, e, Supplementary Fig. 3c). The onset occurs around July with warm SST in the western Pacific and anomalous westerlies and convection in the west of the warm SST (Supplementary Fig. 4g, h). The zonal advective feedback is dominant in the CP El Niño events for both periods during the onset phase (Fig. 1f; 10 of 11 CP El Niño events in these two periods, or 90.9%, are dominated by zonal advective feedback[13]; Supplementary Table 1). The spatiotemporal evolution of EP El Niño events is different from those of extreme and CP El Niño events (Fig. 1g). The initial warm anomalies originate from the EP and then propagate westward. The thermocline feedback is dominant for EP events during the onset phase (Fig. 1h).

To further confirm the similarity of El Niño events that occurred in the 1870s to 1900s and in the 1980s to the present, we compare the climate impacts of El Niño events in the two periods. The composite of boreal winter air temperature anomaly patterns during extreme El Niño events in the period of 1875–1905 closely resembles those in the post-1980s (Supplementary Fig. 5a, b), with a high spatial correlation coefficient of 0.68 ($p < 0.001$). The boreal autumn air temperature anomaly patterns during CP El Niño events that occurred from 1875 to 1905 are also similar to those that occurred in the post-1980s (Supplementary Fig. 5d, e). Meanwhile, the extreme and CP El Niño events exert different impacts on global temperature to EP El Niño events (Supplementary Fig. 5c, f). Note that we compare boreal autumn rather than boreal winter because the temperature anomaly patterns show more prominent differences between CP and EP El Niño events in boreal autumn than in boreal winter[38].

The above results imply that the period of 1875 to 1905 is another extreme and CP El Niño event epoch. Here we use a 2-way table to test the statistical significance (see Methods). The result indicates that the regime shifts in the 1900s and 1980s are both significant at the 99% confidence level (Supplementary Table 2). It suggests that factors other than anthropogenic forcing must play a role in the decadal transition of El Niño types. This result is consistent with a model study showing that variations in El Niño behavior can occur on multidecadal and intercentennial time-scales even with fixed climate forcing[23].

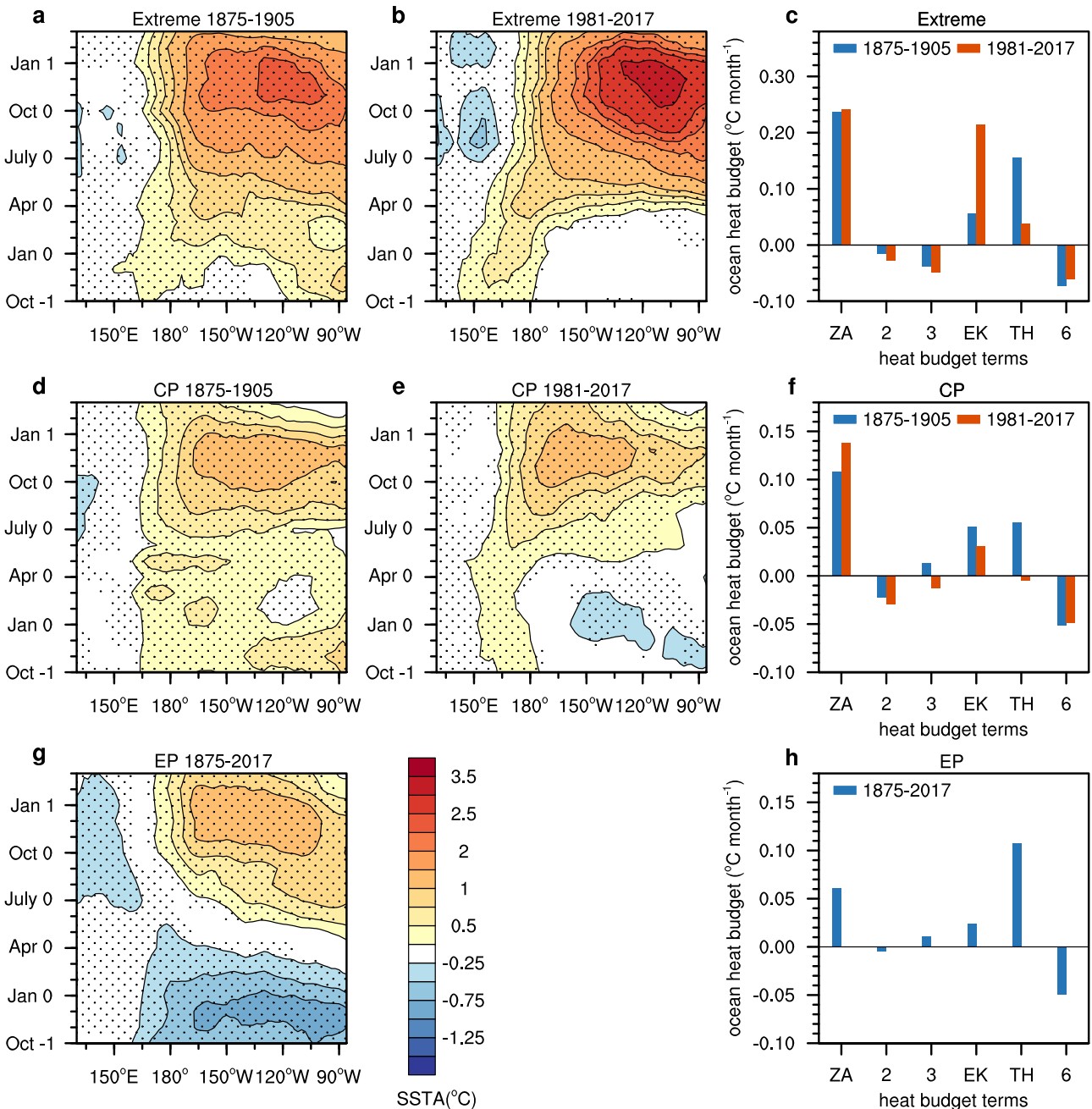

**Fig. 1 | Composite evolution of the equatorial Pacific sea surface temperature (SST) anomalies and heat budget analysis in extreme and Central Pacific (CP) El Niño for two periods. a, b** The evolution of equatorial Pacific averaged (5°S–5°N) sea surface temperature anomalies (SSTA) (shading units: °C) for extreme El Niño events during the period of (**a**) 1875–1905 and (**b**) 1981–2017. **d, e** Same as **a, b** but for CP El Niño events. **g** Same as **a** but for Eastern Pacific (EP) El Niño events during the period of 1875–2017. The stippling indicates the regions where the signal (group mean) is larger than the noise (one standard deviation from the group mean of each member). The group mean and standard deviation are calculated based on the events used in each panel. The anomalies are calculated referenced to the climatology of the full period and linearly detrended. **c, f** Comparison between the period of 1875–1905 and 1981–2017 in the ocean mixed-layer heat budget analysis

of extreme El Niño (**c**) and CP El Niño (**f**) events during their respective onset phases (onset phases are defined as the month when the value of the Niño-3.4 index first exceeds 0.5 °C and the 2 months after that) over the central-eastern Pacific (5°S–5°N, 180°–80°W). **h** Same as **c** but for EP El Niño events during the period of 1875–2017. The terms $-u'\partial\bar{T}/\partial x$, $-\bar{w}\partial T'/\partial z$, and $-w'\partial\bar{T}/\partial z$ denote the zonal advective feedback (ZA), thermocline feedback (TH) and upwelling feedback (EK), respectively. The 6 terms from left to right are $-u'\partial\bar{T}/\partial x$ (bar 1 denoted by ZA), $-\bar{u}\partial T'/\partial x$ (bar 2), $-u'\partial T'/\partial x$ (bar 3), $-w'\partial\bar{T}/\partial z$ (bar 4 denoted by EK), $-\bar{w}\partial T'/\partial z$ (bar 5 denoted by TH), and $-w'\partial T'/\partial z$ (bar 6). The units in the ordinates are °C month⁻¹. The merged HadISST1, ERSST5 and Kaplan SST data from 1871 to 2017 was used. The heat budget is calculated based on the merged Simple Ocean Data Assimilation (SODA) and Global Ocean Data Assimilation System (GODAS) data.

To examine the sensitivity of the extreme and CP El Niño event epoch of 1875-1905 to the traditional classification method, we categorized 38 El Niño into two broad types (EP and CP) based on the Niño method[13,14,39] (see Supplementary Discussion), and then we further divide the EP type El Niño events into moderate EP and extreme El Niño

events by the criterion of DJF-averaged value of the normalized Niño3 index below or above a threshold value of 1.75 standard deviations (s.d.) (see Supplementary Fig. 2e and Supplementary Discussion). Based on the traditional classification method, we identified a total of 16 CP events and 8 extreme events, with 7 CP and 3 extreme events

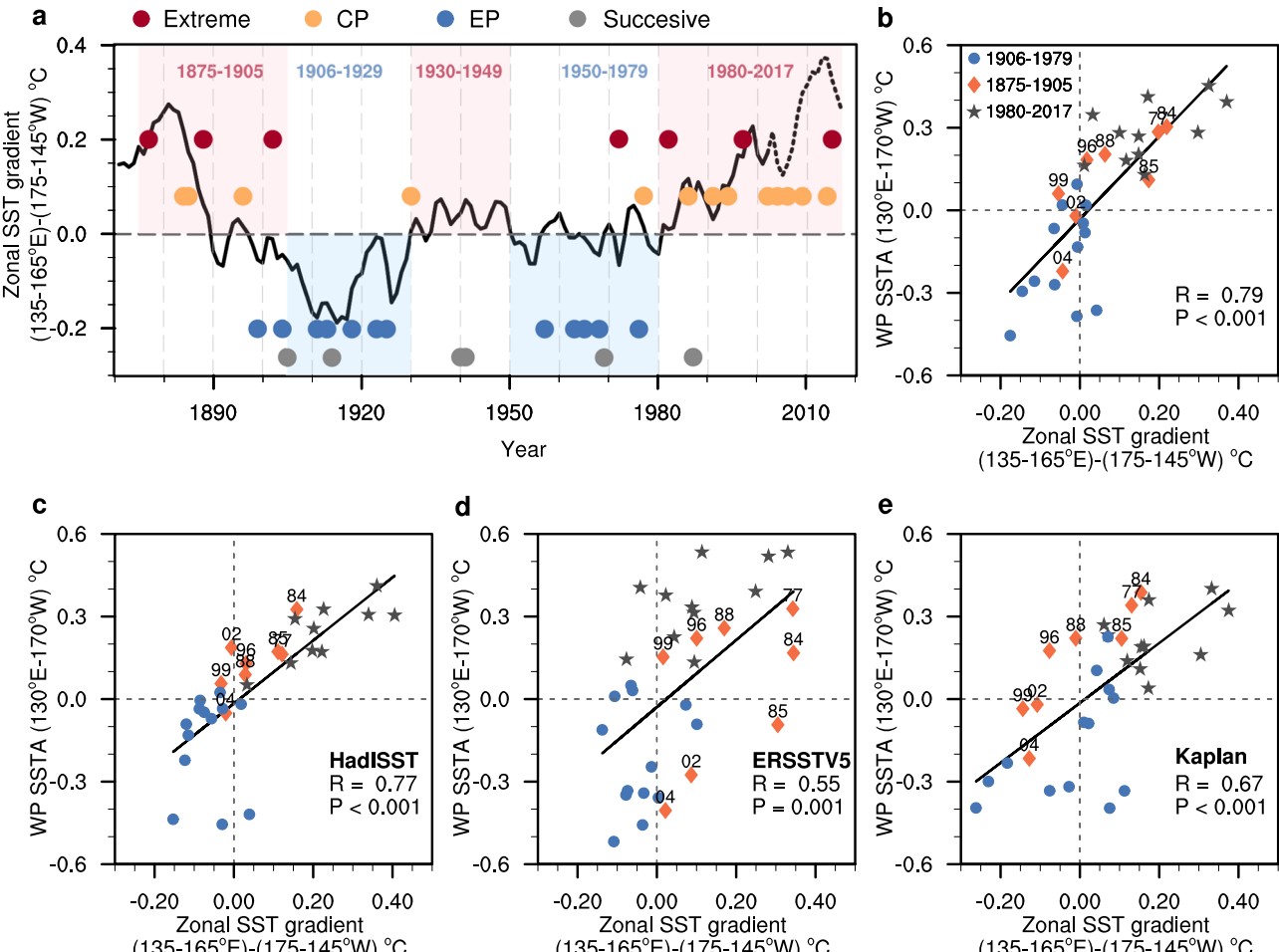

**Fig. 2 | The changing El Niño types from 1871 to 2017 and their relationship with the mean state zonal sea surface temperature (SST) gradient. a** The occurrence of different types of El Niño events. The 38 El Niño events are classified into extreme (red), Central Pacific (CP, orange), Eastern Pacific (EP, blue), and Successive (gray) events. The time series of the 31-year running mean, annual-mean zonal SST gradient [western Pacific SST (135–165°E) minus central Pacific SST (165–145°W)] for the observations from 1871–2017 (°C, relative to the mean of 1901–2010, black line). The positive (black line is greater than the 1901–2010 mean) and negative (black line is less than the 1901–2010 mean) phases of the zonal equatorial SST gradient are represented by light red and light blue shading, respectively. The dashed lines represent not full 31-year running. **b** The relationship between the mean-state zonal

SST gradient and the western Pacific (WP) (130°E–170°W) sea surface temperature anomalies (SSTA) in merged SST during the El Niño onset phase from April (0) to August (0). The merged SST from 1871 to 2017 was used in **a**, **b**. **c**–**e** Similar to **b** but for HadISST (**c**), ERSSTV5 (**d**), and Kaplan (**e**). The black stars and orange diamonds in **b**–**e** represent the El Niño events that occurred from 1875 to 1905 (year 1877, 1884, 1885, 1896, 1899, 1902, 1904) and 1980 to 2017 (year 1982, 1986, 1991, 1994, 1994, 1997, 2002, 2004, 2006, 2009, 2014, 2015), respectively. Solid blue circles represent El Niño events that occurred from 1906 to 1979 (year 1911, 1913, 1918, 1923, 1925, 1930, 1957, 1963, 1965, 1968, 1972, 1976, 1977). The correlation (R) and the P-value of linear regression (black solid line) are also shown. The mean state is defined by the 31-year running mean.

existing in the period from the 1980s to the present and 3 extreme and 3 CP events consecutively occurring in the period 1870s to the 1900s. Thus, the period of 1875–1905, like the recent decades, characterized by increased extreme El Niño and CP El Niño events are still detectable when employing the traditional classification method.

**Decadal variations in El Niño types modulated by the AMO**

What has caused the observed decadal transition of El Niño types? Previous studies suggest a relationship between changes in El Niño types and those in the Pacific zonal temperature gradient[40,41]. Figure 2a shows the relationship between the Pacific zonal temperature gradient and El Niño types using the merged SST data. El Niño behavior is tightly connected with the variation in the zonal mean SST gradient (defined as the 31-year running mean of the difference between SSTA in the western Pacific [135°E–165°E] and SST in the eastern Pacific [175°W–145°W]) on a multidecadal time series: that is, the zonal gradient is stronger in the higher-frequency periods of extreme and CP El Niño events than in the higher-frequency period of EP El Niño events (Fig. 2a).

Observation and model experiments suggest that the enhanced zonal SST gradient could provide a favorable condition for amplifying the zonal advective feedback, and the zonal advective feedback is a major dynamical feedback process, especially in the developing stage of El Niño initiated in the western Pacific[13,40,41]. The enhanced zonal advective feedback over the CP is conducive to triggering the development of El Niño over the western Pacific region[40,42]. We noted that both extreme and CP El Niño events are associated with an initial warm anomaly in the western Pacific, while EP El Niño events are not (Fig. 1; Supplementary Fig. 3). We measure the initial development of El Niño over the western Pacific by calculating the observed western Pacific (130°E–170°W) SSTA during the El Niño onset phase (April through August). The western Pacific SSTA of El Niño is closely related to the mean-state zonal SST gradient (r = 0.79, P < 0.001, Fig. 2b), indicating that extreme El Niño and CP El Niño events rather than EP El Niño events tend to occur during periods of strengthened mean-state zonal SST gradients. This result remains qualitatively unchanged if we use the HadISST1 ERSSTV5, and Kaplan SST datasets

separately (Fig. 2c-e). Our results suggest that zonal SST gradient change is a controlling factor in determining the decadal transition of El Niño types.

Figure 3a shows the correlation of the SST field with the zonal SST anomaly gradient in the merged SST dataset. Here, the signal induced by external forcing, including the change in greenhouse gases (GHGs), natural forcing (NAT), and anthropogenic aerosols (AA), has been removed in the SST field and the zonal SST anomaly gradient (Fig. 3a, see Estimates of the forced and internal components in Method). The spatial correlation field shows significant positive values over the western Pacific, suggesting that the equatorial zonal SST gradient is dominated by the western Pacific. Meanwhile, significant positive values are also found over the North Atlantic Ocean and the pattern resembles the AMO (Fig. 3a), suggesting that the zonal SST anomaly gradient is related to the AMO. This result is consistent with previous studies showing that the AMO could remotely affect the mean state of the Pacific[43–46]. Thus, the AMO seems to be a key candidate for leading to the multidecadal change in El Niño types.

We further analyzed the relationship between the AMO and El Niño diversity based on outputs from the fifth and sixth phases of the Coupled Model Intercomparison Project (CMIP5 and CMIP6). We use outputs from 20 CMIP5 and 23 CMIP6 pre-industrial control simulations over the last 300 years. The CMIP5 and CMIP6 models could reproduce the observed AMO characteristics, with similar spatial patterns in the North Atlantic Ocean (Supplementary Fig. 6). Due to the model's bias in simulating the evolution of El Niño, we use the El Niño onset phase (April to August) averaged western Pacific SSTA > 0 °C to roughly distinguish the extreme/CP from EP events. A total of 30 of 43 CMIP5 and CMIP6 models (70%) simulated an increased western Pacific SST for the AMO-positive state minus the AMO-negative state (Fig. 3b). The increased western Pacific SST enhances the zonal SST gradients, which is conducive to the development of El Niño in the Niño4 region[13,40]. Models that generate a larger increase in the western Pacific SST/zonal SST gradient for the AMO-positive state minus the AMO-negative state tend to simulate a larger increase in the frequency of extreme/CP El Niño events for the AMO-positive state minus the AMO-negative state ($r = 0.66$, $P < 0.001$). A total of 29 of 43 CMIP5 and CMIP6 models (67.4%) produce an increased occurrence of CP and extreme El Niño for the AMO-positive state minus the AMO-negative state.

We also use multicentury palaeoclimate reconstructions to examine the relationship between the AMO and the decadal modulation of the different types of El Niño events prior to the instrumental record. The palaeoclimate reconstructions include a 700-year (1300–2006) El Niño Niño3.4 index reconstruction[20], a 1200-year (800–2008) AMO index reconstruction[47], and a 400-year (1617–2008) record of CP El Niño events reconstructed from ENSO-sensitive proxy records[15]. To focus on the decadal modulation of different types of El Niño events, we count the occurrences of extreme El Niño events (defined by a Niño-3.4 index >1.2 s.d.) and CP Niño events (identified following the pioneering work of ref. [15]) over a 21-year sliding period. The 21-year sliding frequencies of extreme El Niño events and CP Niño events increased in the positive phase of the AMO (Fig. 3c) and they are statistically significant above the 95% confidence level according to a bootstrap test (see Methods).

How does AMO impact El Niño diversity? We examined the process using the 5 CMIP6 and 3 CMIP5 models, in which the western Pacific SST responses to the AMO are comparable to the observations (Supplementary Fig. 7a). The simulated spatial pattern of the Pacific response to the AMO among these models exhibits a common feature: annual mean anticyclonic flows over the Northwest Pacific (NWP), significant warm SSTA over the western Pacific and the subtropical North Pacific (SNP), and strong northward flows from the tropics towards the SNP (Supplementary Fig. 7). We find that these features of the Pacific response to AMO derived from CMIP5 and CMIP6 are consistent with those derived from the observations (Supplementary Fig. 8) and a suite of Atlantic Pacemaker experiments[45]. Such a spatial pattern of the Pacific response implies that the AMO could induce anomalous high pressure over SNP (Supplementary Fig. 8b), thereby causing SNP warming via wind–evaporation–SST feedback, and SNP warming could further develop warm SSTs in the western Pacific through SST–sea level pressure–cloud–longwave radiation positive feedback[45]. The zonal SSTA gradient over the central Pacific increases as the western Pacific SSTA increases (Supplementary Fig. 9b), which in turn leads to enhanced zonal advection feedback in the development of El Niño (Supplementary Fig. 9c). Thus, a positive AMO corresponds to a warm SSTA in the western Pacific, a large zonal SSTA gradient over the central Pacific, and an enhanced CP and extreme El Niño, vice versa. These results imply that the AMO could modulate the frequency of the three types of El Niño events, with more frequent extreme and CP events in the positive phase of the AMO.

## Contribution of external forcing and internal variability

The zonal SST gradient is divided into the externally forced component and internal variability-related component in Fig. 4a (see Estimates of the forced and internal components in Method). The externally forced (EX) zonal SST gradient shows an upward trend (Fig. 4a). In response to external forcing, the eastern Pacific warms less than the western Pacific due to upwelling and the shallow thermocline, strengthening the zonal SST gradient via Bjerknes feedback[48,49]. The contribution of EX to the multidecadal equatorial zonal SST gradient variations increased over time and exceeded that of internal variability from 1981 to 2017 (Fig. 4b).

Although the contribution of EX to the multidecadal equatorial zonal SST gradient variations varies in 5 periods, there is a significant linear correlation between the mean zonal SST gradient and the percentages of occurrence ratio of different types of El Niño events to total El Niño events (Supplementary Fig. 10). We use a linear regression model to estimate the EX-induced and IV-induced occurrence of different types of El Niño events from 1981 to 2017 (see Methods). Note that the results of the linear regression for the sum of extreme and CP El Niño events (Supplementary Fig. 10a) are consistent with the model result (Fig. 3b), providing high confidence for the estimation. If both EX and IV are considered in the model, we use the observed zonal gradient in the last 40 years as a predictor in the linear regression model, which yields ~9.1 extreme and CP El Niño events (including ~2.7 extreme and ~6.3 CP El Niño events) and ~0 EP El Niño events in the last 40 years, which are close to the actual number of occurrences of El Niño events (Fig. 4c). If only IV was considered in the model, ~1.8 extreme and ~4.1 CP El Niño events are predicted, and the AMO accounts for 77.8% and 75.6% of IV-induced extreme and CP El Niño events, respectively. This result suggests that the extreme and CP El Niño event epochs still exist even without considering anthropogenic forcing. Moreover, anthropogenic forcing also plays a role in changing the El Niño diversity in the last 40 years. The linear regression model result shows that anthropogenic forcing-induced warming accounts for up to ~1 more extreme and ~2 more CP El Niño events from the 1980s to the present. Our result, that the recent changes in El Niño events were attributed to synchronized effects of greenhouse warming and internal variability, still holds when the traditional El Niño classification method is used (Supplementary Information gives a detailed analysis).

## Discussion

In this study, we found an extreme El Niño and CP El Niño epoch from 1875 to 1905, with the El Niño events that occurred in this period showing similar spatiotemporal evolution, dynamic processes, and global climatic impacts to those that occurred after 1980, suggesting a role of internal variability. Then, we revealed that anthropogenic forcing and the internal variability associated with the AMO synchronously contributed to the increased frequency of extreme and CP events since

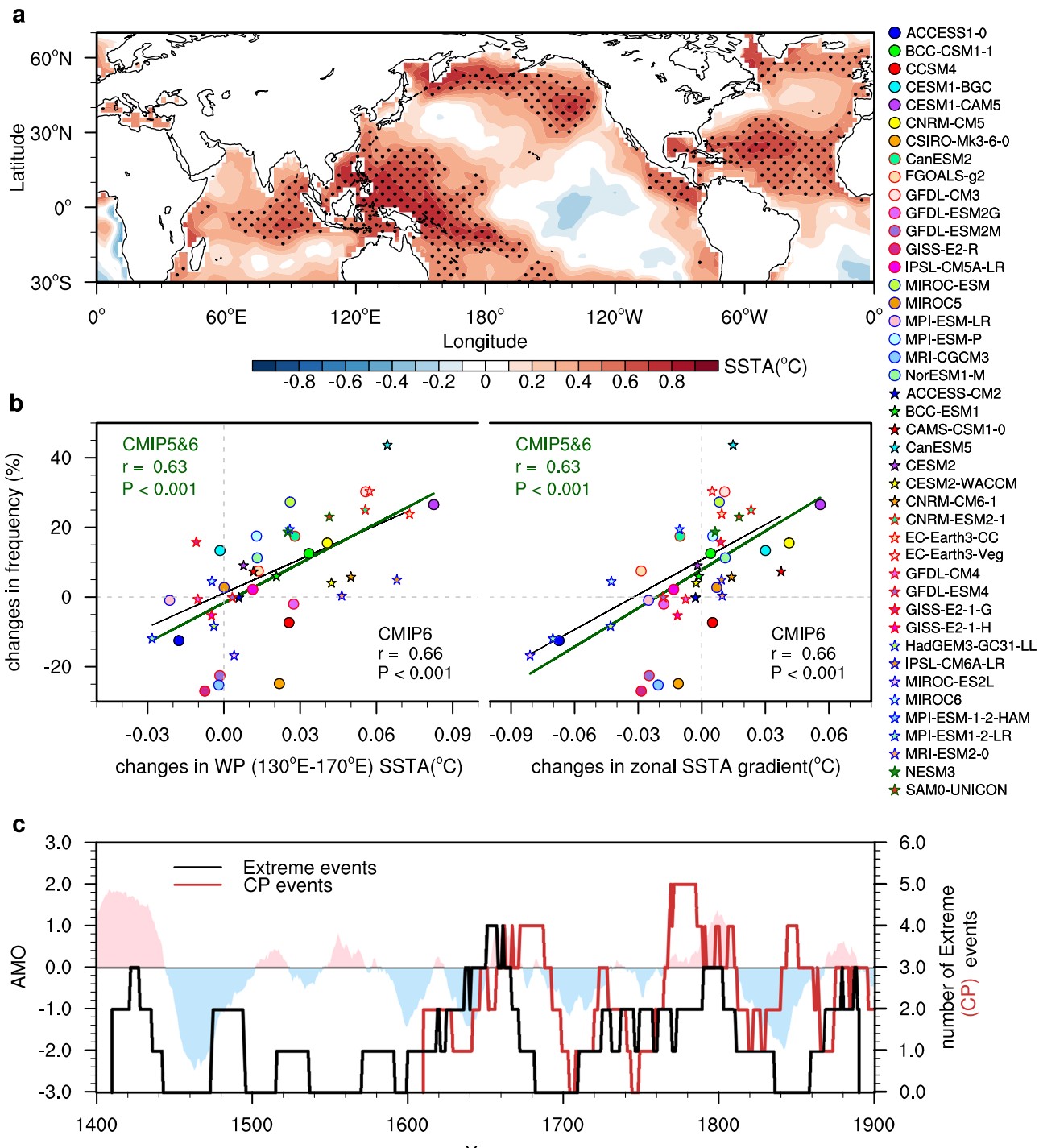

**Fig. 3 | The relationship between the Atlantic Multidecadal Oscillation (AMO) and El Niño types. a** The correlation between the internal variability (IV) induced zonal sea surface temperature (SST) anomaly gradient and the residual SST anomaly (SSTA) field. The black stippling indicates statistically significant correlations at the 0.05 level. The merged SST from 1871 to 2017 was used. **b** Inter-model relationship between the changes in the 21-year running mean western Pacific (WP) SSTA (130°E–170°E) /Zonal SST gradient [WP SST (155°E–175°W) minus central Pacific SST (115°W–145°W)] (x-axis, units: °C) and the frequency changes in the extreme/ Central Pacific (CP) El Niño events (y-axis, units: %) and for AMO positive state (AMO-positive state) minus AMO-negative state. Due to the model's bias in simulating the evolution of El Niño, we use the El Niño onset phase (April to August) averaged WP SSTA > 0 °C to roughly distinguish the extreme/CP from Eastern Pacific (EP) events. 20 CMIP5 and 23 CMIP6 pre-industrial control simulations were used. **c** The 21-year sliding frequency of extreme El Niño events (black solid line)/CP El Niño events (red solid line) in the reconstruction and the normalized 21-year running mean AMO reconstruction index[47]. The 21-year sliding frequency is defined by counting extreme/CP El Niño events during the 21 years. The CP El Niño events in the reconstruction are according to ref. [15]. The extreme El Niño events are defined by Niño-3.4 reconstruction index >1.2 standard deviations.

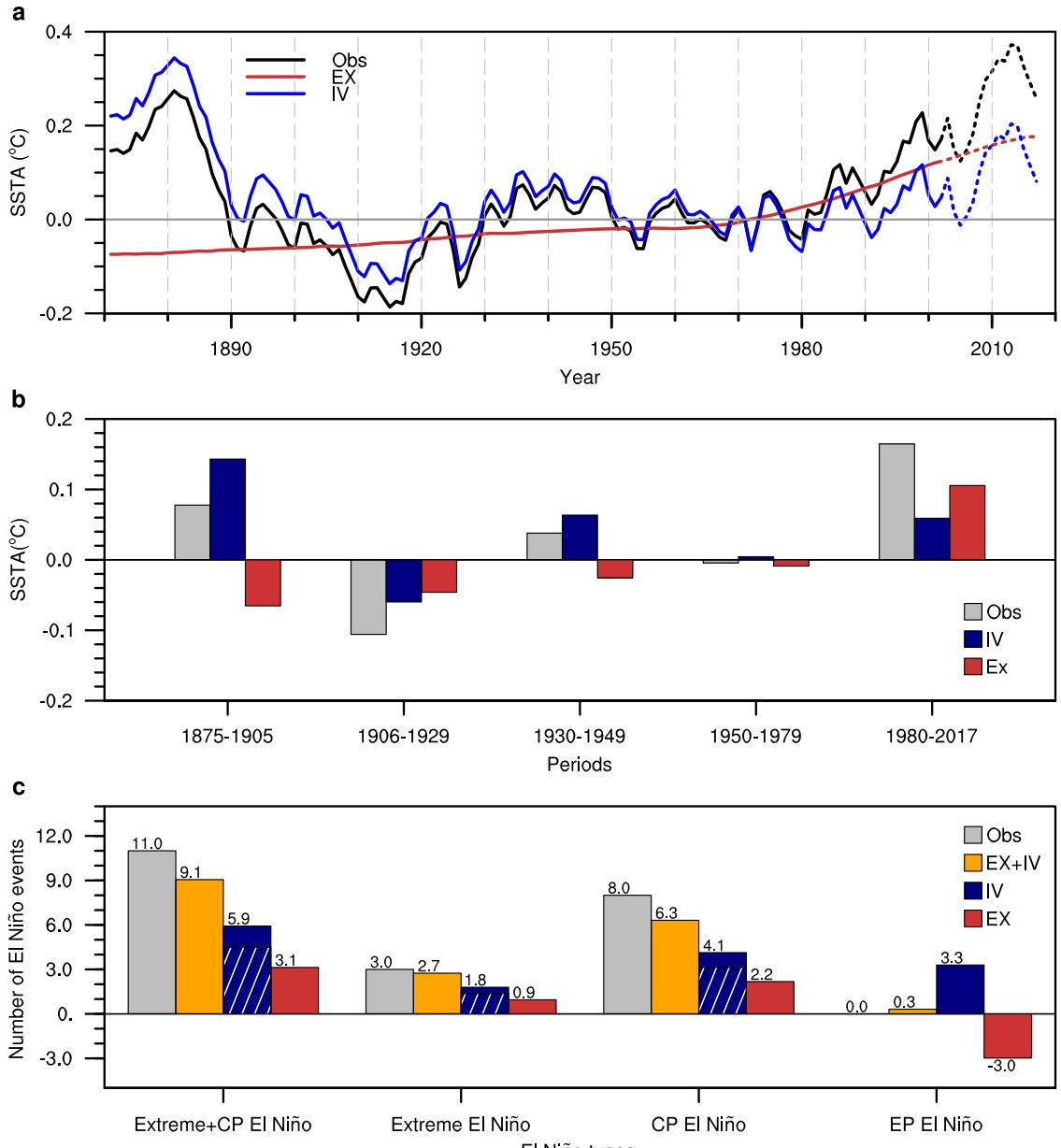

**Fig. 4 | Contributions of external forcing and internal variability to the frequencies of extreme El Niño events and central Pacific El Niño events over the past four decades. a** Time series of the 31-year running mean, annual-mean zonal sea surface temperature anomaly (SSTA) gradient [western Pacific SSTA (135–165°E) minus central Pacific SSTA (165–145°W)] for the observations from 1871–2017 (Obs, black line, the same as the black line in Fig. 2a, relative to the mean of 1901–2010), the estimated externally forced signal (EX, red) and internal variability (IV, dark blue) in the observations. The dashed lines represent not full 31-year running. **b** Time-averaged zonal equatorial SST anomalies (°C, relative to 1901–2010 mean) gradient from Obs (gray), IV (blue), and EX (red) for 5 different periods marked in Fig. 2a. The merged sea surface temperature (SST) from 1871 to 2017 was used. **c** Histograms for the estimated numbers of different types of El Niño events in 1981–2017 based on a linear regression model (see Methods and Supplementary Fig. 10). The orange, blue, and red bars denote the EX + IV-, IV-, and EX-induced numbers of different types of El Niño events, respectively. Slant hatching denotes the Atlantic Multidecadal Oscillation (AMO) induced numbers of different types of El Niño events.

1980 by modulating the background warming in the western Pacific and the associated zonal advective processes. Furthermore, the internal variability is estimated to contribute to the observed extreme and CP El Niño event epochs since 1980 by up to 65%, with AMO accounting for more than 3/4 of that. However, anthropogenic forcing can explain ~1 more extreme and ~2 more CP events in the last 40 years. Our findings highlight that both the external forcing and the internal variability of the AMO are important factors to be considered in projecting the diversity of El Niño events in a changing climate.

Future changes in El Niño are a crucial issue. Climate models project a tropical Pacific mean state change under global warming,

which could not only exert a significant impact on tropical precipitation patterns[50,51] but also influence the ENSO characteristics[52,53]. As extreme El Niño and CP El Niño events tend to occur in periods of enhanced equatorial west-minus-east SST gradient, the projection of El Niño types should partly depend on the projection of this gradient in the Pacific.

## Methods
### Observation datasets
We analyze (a) the Hadley Center Sea Ice and SST dataset version 1 (HadISST1[31]) (b) the Extended Reconstructed Sea Surface Temperature

(ERSST) version 5 global SST monthly dataset[32] and (c) the Kaplan extended SST version 2 dataset[33] for 1871–2017. Data uncertainties exist prior to 1950 due to the sparse coverage of instrumental observations across the equatorial Pacific[54]. To reduce the uncertainty in the three SST datasets, we used a "merged monthly mean SST dataset (merged SST hereafter) following the pioneering work of ref. [34]". The merged SST is defined as the arithmetic mean of the three monthly mean SST datasets. In addition to the "merged" SST dataset, we also used the HadISST1, ERSSTV5, and Kaplan SST datasets to examine the sensitivity of the results to different instrumental datasets (Supplementary Fig. 2). Although slight differences still exist, our key conclusion is not sensitive to the SST products, and it also provides some confidence to our result.

The surface winds and atmospheric circulation fields are derived by merging the NOAA-CIRES Twentieth Century Reanalysis (20CRv2c)[55] from 1871 to 2012 and NCEP/DOE Reanalysis 2 data[56] from 1979 to 2018. To ensure temporal consistency, we use the differences in monthly climatology between 20CRv2c and NCEP/DOE2 data during the overlap period 1979–2012 to calibrate the mean state of NCEP/DOE. The ocean reanalysis dataset used is from SODA version 2.2.4 for 1871–2008[57] with a resolution of 0.5° × 0.5° and extended from 2009 to 2018 with the Global Ocean Data Assimilation System (GODAS) with a 1° × 1° grid[58]. We use the difference between GODAS and SODA2.2.4 during the overlay period 1980–2008 to calibrate the mean state of GODAS. The land air temperature used is the HadCRUT5[59] gridded monthly surface temperature datasets. The long-term ship-observed sea level pressure (SLP) and marine cloud cover were used from the International Comprehensive Ocean–Atmosphere Data Set (ICOADS) release 3[60]. The SLP field derived from the Hadley SLP (HadSLP2) dataset[61] was also used.

### Reconstruction data

We also use multi-century palaeoclimate reconstructions to extend the observational record. The Niño-3.4 index and AMO index reconstructions used are from the NOAA/World Data Center for Paleoclimatology archive, including a 700-year (1300–2006) El Niño Niño-3.4 index reconstruction based on 2,222 tree-ring chronologies from both the tropics and mid-latitudes in both hemispheres[20] and a 1200-year (800–2008) Atlantic Multidecadal Variability/Oscillation (AMV/AMO) Reconstructions based on 46 annual resolved terrestrial proxy records[47]. A 21-year running mean is applied to the AMV/AMO reconstruction index. The first and last 100 years are cut-offs to minimize edge effects introduced by the spectral filter. The overlay period of 1401–1900 is used. We define extreme El Niño as when the Niño 3.4 index is greater than 1.2 standard deviations. In addition to the multi-century record of extreme El Niño events reconstructed from the Niño-3.4 reconstruction, we also use 400 years (1601–2008) of CP El Niño events reconstructed from 27 seasonally resolved coral records[15] to study the relationship between CP events and AMO. The period of 1601–1900 is used, and a total of 34 CP events are identified based on the pioneering work of ref. [12]: 1618, 1620, 1641, 1652, 1657, 1667, 1672, 1677, 1682, 1688, 1693, 1718, 1730, 1733, 1759, 1769, 1775, 1778, 1779, 1781, 1790, 1799, 1801, 1808, 1816, 1832, 1840, 1850, 1853, 1854, 1873, 1884, 1885, and 1895 years.

### Model simulation data

We used all forcing (ALL) and single-forcing experiments forced separately by greenhouse gases (GHGs), natural forcing (NAT), and anthropogenic aerosols (AAs) from 27 CMIP6 models (Supplementary Table 3). We used the pre-industrial control runs from 23 CGCMs of CMIP6 and 20 CGCMs of CMIP5. The 23 CMIP6 models were ACCESS-CM2, BCC-ESM1, CAMS-CSM1-0, CanESM5, CESM2, CESM2-WACCM, CNRM-CM6-1, CNRM-ESM2-1, EC-Earth3-CC, EC-Earth3-Veg, GFDL-CM4, GFDL-ESM4, GISS-E2-1-G, GISS-E2-1-H, HadGEM3-GC31-LL, IPSL-CM6A-LR, MIROC-ES2L, MIROC6, MPI-ESM-1-2-HAM, MPI-ESM1-2-LR,

MRI-ESM2-0, NESM3, and SAM0-UNICON. The 20 CMIP5 models were ACCESS1-0, BCC-CSM1-1, CCSM4, CESM1-BGC, CESM1-CAM5, CNRM-CM5, CSIRO-Mk3-6-0, CanESM2, FGOALS-g2, GFDL-CM3, GFDL-ESM2G, GFDL-ESM2M, GISS-E2-R, IPSL-CM5A-LR, MIROC-ESM, MIROC5, MPI-ESM-LR, MPI-ESM-P, MRI-CGCM3, and NorESM1-M. These models were usually used to understand the changing behavior of El Niño[62,63].

### Definition of El Niño years

We define El Niño events based on SST anomalies averaged in the Niño-3.4 region (5°N–5°S, 120°W–170°W), here called the Niño-3.4 index. We denote the El Niño first developing year, the following year, and the year prior to El Niño as year 0, year 1, and year −1, respectively. We smooth the Niño-3.4 index with a 3-month running-mean filter. El Niño events are defined as occurring when the detrended October–February (ONDJF)-averaged Niño-3.4 index is greater than or equal to 0.6 °C. A total of 38 El Niño events are identified using the merged SST dataset from 1871 to 2017.

### Cluster analysis

We used a nonlinear K-means cluster analysis focused on the spatio-temporal evolution of El Niño events from the onset to the mature phase (from Nov-1 to Oct 0), which is depicted by SST anomalies averaged between 5°S and 5°N. The data used in the cluster analysis were 3-month running mean SST anomalies based on the original SST data, and to focus on the onset phase to mature phase, only SSTAs greater than 0.3 °C were used for the K-means cluster analysis, which is different from the pioneering work of ref. [34].

In the K-means cluster analysis, we use the squared Euclidean distance to measure the similarity between each cluster member and the corresponding cluster pattern. The silhouette clustering evaluation criterion was used to evaluate the performance of the cluster analysis. A high silhouette value indicates that the member is well matched to its cluster and poorly matched to neighboring clusters. Through nonlinear K-means cluster analysis of the evolution of El Niño events, we obtained four types of El Niño events. Supplementary Fig. 1 shows the silhouette values for each El Niño event within each of the 4 clusters for the 1871–2017 period for the merged SST, HadISST1, ERSSTV5, and Kaplan SST datasets. Furthermore, we tested different SST datasets to identify the extreme and CP El Niño events, and they yielded similar results, indicating that the results are robust and not sensitive to the exact choice of dataset (see Supplementary Discussion). Our results are not sensitive to the classification method—for example, we find similar results if we used the traditional Niño method[13,14,39] to categorize El Niño events into EP and CP events and then extracted the extreme El Niño events (see Supplementary Fig. 2, Supplementary Fig. 10-11 and Supplementary Discussion).

### Ocean mixed layer heat budget equation

To compare the dominant dynamic processes of extreme and CP types of El Niño between the two periods, we conduct an ocean mixed layer heat budget analysis. The heat budget is computed according to the following equation:

$$\frac{\partial T'}{\partial t} = \left\langle -u'\frac{\partial \bar{T}}{\partial x} \right\rangle + \left\langle -\bar{u}\frac{\partial T'}{\partial x} \right\rangle + \left\langle -u'\frac{\partial T'}{\partial x} \right\rangle + \left\langle -v'\frac{\partial \bar{T}}{\partial y} \right\rangle + \left\langle -\bar{v}\frac{\partial T'}{\partial y} \right\rangle + \left\langle -v'\frac{\partial T'}{\partial y} \right\rangle$$
$$+ \left\langle -w'\frac{\partial \bar{T}}{\partial z} \right\rangle + \left\langle -\bar{w}\frac{\partial T'}{\partial z} \right\rangle + \left\langle -w'\frac{\partial T'}{\partial z} \right\rangle + \frac{Q'_{net}}{\rho C_p H} + R \tag{1}$$

where $T$ is the mixed layer temperature; $u$, $v$, and $w$ represent the zonal, meridional, and vertical ocean currents, respectively; $Q_{net}$ is the net surface heat flux comprised of shortwave, longwave, latent, and sensible heat fluxes; $\rho$ (=$10^3$ kg m$^{-3}$) is the ocean water density; $C_p$ (=4000 J kg$^{-1}$ K$^{-1}$) is the specific heat of water; $H$ is the climatological mixed layer depth as a constant 50 m; $R$ represents the residual term; and the overbar and prime denote the climatological mean and

anomalies, respectively. The analysis result is not sensitive to the constant H we choose, such as H = 70 m. $-u'\partial\bar{T}/\partial x$, $-\bar{w}\partial T'/\partial z$, and $-w'\partial\bar{T}/\partial z$ denote the zonal advective feedback, thermocline feedback and upwelling feedback, respectively.

### Contingent (2-way) table and Pearson's chi-square test

The contingency table, also known as a cross-tabulation or a two-way table, is usually used to present categorical data in terms of frequency counts. We use Pearson's chi-square test to examine independence between the row and column variables in the contingency table[64]. Pearson's chi-square test statistic follows an asymptotic chi-square distribution with (r−1)X(c−1) degrees of freedom when the row and column variables are independent. It is calculated as

$$\chi^2 = \sum_{i=1}^{r}\sum_{j=1}^{c}\frac{\left(O_{ij}-E_{ij}\right)^2}{E_{ij}} \qquad (2)$$

where $O_{ij}$ is the observed value shown in the contingency table and $E_{ij}$ is the expected value. $E_{ij}$ is calculated as $E_{ij}=\frac{N_iN_j}{N}$, $N_i=\sum_{j=1}^{c}O_{ij}$, and $N_j=\sum_{i=1}^{r}O_{ij}$. where r and c are the numbers of rows and columns in the contingency table, $N_i$ is the row total and $N_j$ is the column total.

### Estimates of the forced and internal components

Many previous studies used the global-mean SST (GMSST) from the multi-model ensemble mean (MMM) simulation to represent the externally forced signal. It is a preferred way to define the forced signal, which has the same temporal evolution over all grid boxes because it is determined by the external forcing series[65]. This method is based on the principle that the internal variations among the ensemble runs are usually uncorrelated. Thus, after averaging over a large number of ensemble simulations, the uncorrelated internal variations among the ensemble runs were largely smoothed out[65]. To calculate the MMM, we averaged all the ensemble runs for each single-forcing simulation from all the models. The MMM was similar if we first average over the available ensemble runs for each model to derive the ensemble mean for each model and then average the ensemble mean over all the models with the same weight for each model. Then we calculate the GMSST time series, $T_{GHG}, T_{NAT}, T_{AA}$ from the MMM of GHG, NAT and AA forcing, respectively. A combined index $aT_{GHG}+bT_{NAT}+cT_{AA}$ is constructed based on a multiple linear regression of the observed GMSST time series onto $T_{GHG}, T_{NAT}, T_{AA}$, where $a,b,c$ are the regressed coefficients. We use the combined index to estimate the externally forced component of the observed GMSST. We removed changes associated with the externally forced component through linear regression from observed SSTs at each grid point. The residual SST fields primarily contain unforced internal variations[65].

### Linear regression method

Since there is a good relationship between the background zonal SSTA gradient and the frequency of different types of El Niño occurrence, for example, frequent occurrence of extreme and CP El Niño events is observed in the period of 1875–1905, 1930–1949 and 1981–2017 coincide with the enhanced zonal SST gradient, while in the other period (from 1905 to 1930 and from 1950 to 1980), EP El Niño events are dominant with a weakened zonal SST gradient. Here, we attempt to estimate the relative contribution of EX- and IV-induced zonal gradients to the El Niño type changes in the last 40 years. We use the background zonal SSTA gradient as predictor $X$ and the occurrence ratio of the sum of extreme and CP El Niño events to total El Niño events in one period as response variable $Y$

$$Y = \beta_0 + \beta_1 X \qquad (3)$$

We used the merged SST dataset shown in Fig. 2a and the HadISST1, ERSSTv5, and Kaplan SST datasets shown in Supplementary Fig. 2, and for each dataset, we calculated the mean zonal SSTA gradient in five periods (1875–1905, 1906–1929, 1930–1949, 1950–1980, 1981–2017) and the occurrence ratio of the sum of extreme and CP El Niño events to total El Niño events in the five periods, so we obtained N = 20 and used them to calculate the intercept $\beta_0$ and slope $\beta_1$. Then, we calculate the IV-induced and IV + EX-induced mean zonal SSTA gradients from 1981 to 2017 and use them to estimate the IV-induced and IV + EX-induced occurrence ratio of the sum of extreme and CP El Niño events to total El Niño events. A total of 11 El Niño events occurred during the period 1981 to 2017, so we could further estimate the IV-induced and IV + EX-induced occurrence of the sum of extreme and CP El Niño events in this period. The difference between IV + EX-induced and IV-induced occurrence of total extreme and CP El Niño events in this period is used to estimate the EX-induced occurrence of total extreme and CP El Niño events. We also applied the above analysis to extreme, CP, and EP El Niño events.

### Statistical analyses

Statistical significance tests were based on Student's t test with reduced degrees of freedom[66]. We calculate the effective degrees of freedom (EDF) using the following equation:

$$\frac{1}{N^*} \approx \frac{1}{N} + \frac{2}{N}\sum_{j=1}^{N}\frac{N-j}{N}\rho xx(j)\rho yy(j) \qquad (4)$$

where $N$ is the sample size, and $\rho xx(j)$ and $\rho yy(j)$ are the autocorrelations of the two sampled time series x and y at time lag j. Here, we follow ref. [67] and use the above equation but without the weighting function $(N - j)/N$.

We use a bootstrap test to examine whether the number of extreme El Niño/CP El Niño events is statistically different in the positive phase of AMO and negative phase of AMO. We use Monte Carlo bootstrapping to estimate the probability density function (PDF) of the number of the 21-year sliding frequencies of extreme El Niño events and the number of the 21-year sliding frequencies of CP El Niño during positive and negative phases of the AMO. The resampling procedure was repeated 10 000 times. The 2.5 and 97.5% rankings from the probability distribution function indicate the 95% confidence level. The number of extreme El Niño increased in positive phase of AMO compared to negative phase of AMO is statistically significant above the 95% confidence level if distributions of number of extreme El Niño during positive phase of AMO are well-distinguished from that during the negative phase of AMO.

## Data availability

The data that support the findings of this study are freely available. Observational SST datasets can be acquired from: (1) https://www.metoffice.gov.uk/hadobs/hadisst/ for HadISST1, (2) https://psl.noaa.gov/data/gridded/index.html for ERSSTV5 and Kaplan SST. The surface wind and atmospheric circulation fields observations or reanalysis can be acquired from https://psl.noaa.gov/data/gridded/index.html. The ocean reanalysis dataset can be acquired from: (1) http://apdrc.soest.hawaii.edu/datadoc/soda_2.2.4.php for SODA2.2.4, (2) https://psl.noaa.gov/data/gridded/data.godas.html for GODAS. The HadCRUT5 is available at https://www.metoffice.gov.uk/hadobs/hadcrut5/. The HadSLP2 and ICOADS are available at https://psl.noaa.gov/data/gridded/index.html. CMIP5 and CMIP6 data can be acquired from https://esgf-node.llnl.gov/projects/esgf-llnl/. The reconstruction data can be acquired from http://www.ncdc.noaa.gov/paleo/.

## Code availability

The source code for kmeans clustering used in this study are available at website: https://www.mathworks.com/help/stats/kmeans.html. The

data in this study is mainly analyzed with NCAR Command Language (NCL; v6.6.2, https://www.ncl.ucar.edu/), which is a public access software. Key codes can be accessed from https://doi.org/10.6084/m9.figshare.21546912.v5.

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

## Acknowledgements

The work was supported by the National Natural Science Foundation of China (42141019, 41831175, 41721004, 42175040), STEP (2019QZKK0102), Strategic Priority Research Program of Chinese Academy of Sciences (XDA20060500), the National Basic Research Program of China (2018YFA0605900, 2019YFA0606703). The work was also supported by the Youth Innovation Promotion Association of CAS (2021072), the Chinese Jiangsu Collaborative Innovation Center for Climate Change, the Frontiers Science Center for Critical Earth Material Cycling of Nanjing University, and High-Performance Computing Centers of Nanjing University.

## Author contributions

Q.L. and G.H. designed the research, provided comments, and revised the manuscript. R.G. performed the analysis and drafted the manuscript. K.H. helped organize and revise the draft. X.L. gave comments and contributed to the discussion of the results. R.G., Q.L., G.H., K.H. and X.L. contributed to scientific interpretations and subsequent revisions.

## Competing interests

The authors declare no competing interests.
