## [Peer Review File · Nature Communications]

Greenhouse warming and internal variability synergistically increase extreme and central Pacific El Niño frequency since 1980Reviewers' Comments:

Reviewer #1:

Remarks to the Author:

A review of "Synchronize effects of greenhouse warming and internal variability have increased the frequency of extreme and central Pacific El Niño since 1980" by Ruyu Gan et al.

Key Results

The authors are asking a very clear question – if a change in El Niño properties to more extreme and more Central Pacific (CP) events in the recent decades is due to natural variability or is it externally forced. They complete a cluster analysis of SST datasets to classify El Niño events and to compare similarities in events across time periods. Their main findings suggest there was another time period – 1875-1905 – with increased extreme and CP El Niño events, like the recent decades, but when anthropogenic CO₂ forcing was relatively lower. They then look CMIP6 climate models to investigate the internal variability influence on the decadal observations in changing El Niño properties. They find that the historical time period, as well as recent decades, coincides with a positive phase of the Atlantic Multidecadal Oscillation (AMO). Using a linear regression model, they estimate that internal variability (AMO) contributed to ~65% of the recent increasing extreme and CP El Niño events over the past four decades, with the anthropogenic forcing contributing to the remaining 35%.

Validity

The authors seem to have a reasonable interpretation of the data based on the results of the analyses performed. However, their results should be considered in more context alongside previous studies – particularly studies that have previously suggested AMO may modulate ENSO and paleoclimate studies that reconstruct ENSO pre-1950's (see a few suggested references below).

Significance

The results are highly significant to the field. There is growing evidence that El Niño events will not only become more extreme more frequently under future enhanced greenhouse gas warming, but that they are already observed to be more frequent in the recent decades. However, it's significant they highlight that natural variability still plays a role and can enhance the anthropogenic forcing. Additionally, the authors highlight another period where CP events were more common, similarly to today.

Data and methodology

The authors use three main SST datasets – the HadISST1, ERRST, and Kaplan. The reliability of these SST datasets for the equatorial Pacific reduces beyond 1950 despite the authors claim that they are reliable for the past 150 years (line 250). The authors state they create a "merged SST" dataset to reduce uncertainty. However, the paper gives little attention to uncertainties that may arise, affecting their pre-1950 data (see Deser et al., 2009). In the supplemental material, they complete a sensitivity test to illustrate that the classifications of El Niño events and the number of extreme El Niños of each SST dataset are identical to the merged SST product. However, this still falls short of assessing the uncertainty related to limited data in the region of interest.

Suggested improvements

1. Give more attention to the uncertainty in the SST products and discuss the implications of uncertainties for the results.
2. Put results in context of previous studies. In the "References" section below, are a few suggested references that should be considered in the context of the results. Specifically, there seems to be little regard to previous ENSO-AMO and paleoclimate studies that have tried to address similar questions. For example, Line 35 states "... to the best of our knowledge, no study has done this work to date." is not true. There is a plethora of paleoclimate studies that have tried to answer this question, and some

that come to a different conclusion. Another example, Line 131 states “we suspect that the interdecadal variation in El Niño types was modulated by the AMO..” but does not reference any previous literature that already has investigated this.

3. Improve overall clarity of writing. Below in the “Clarity and context” are more specific suggestions. The writing needs to be more specific with respect to datasets and methods used as it was not always clear what was being done or what datasets were being used. Sentences could be less wordy and more direct. For example, outline more specifics of the study design in the introduction. The results seem out of context until reading the end METHODS section.

4. Improve overall organization of paper. The organization follows the authors’ logical thoughts, which is not necessarily the best way to present work (for examples – Line 132 – “To confirm this suspicion...” and Line 151 – “A natural question follows...”). The paper could be improved with describing what was done, what was found, and then discussing those results into context of other work, within each of the main sections. For example, Line 109 – “Here we show....” is presented before the authors state what they do. Consider walking the reader through the analyses and then end with the interpretation of how El Niño is tightly connected to zonal mean SST.

Clarity and context

Overall, this paper could improve in clarity – in terms of both content and writing. It’s not always clear what the authors are referring to or what datasets they are using for the analyses. Additionally, most sentences are plagued with wordy passive tense and could be improved with a more active tense approach.

Below are specific suggestions to improve clarity:

5. Line 19 – disaster and disastrous are in the same since which is redundant; maybe change to “severe climatic disruptions...”

6. Line 22 – “being located” is awkward; remove and just say, “which is characterized by the peak ocean warming in the central equatorial Pacific...”

7. Line 24 – “being located” is awkward; remove “being” and say “.., SST anomalies located in the far eastern equatorial Pacific...”

8. Line 35 – To the reader it is unclear what the authors are referring to here with “records”. Are they looking at paleo records? Or instrument SST records? Just in the equatorial Pacific? Or at other regions with teleconnections? What do they mean by “using multiple long-term instrumental SST datasets”? I think a better way to clarify that they are looking at SST products, the authors could rephrase to “we examine multiple long-term instrumental SST datasets of El Niño...”

9. Line 46 – Define what datasets the cluster analysis applied to.

10. Line 48 – This statement is confusing as they list the cluster analysis years from 1871-2017 but say it’s the period “not affected much by anthropogenic activities”. Suggest rewording for clarity.

11. Line 51 – Confusing why the classifications were completed for 1901-2017, and not 1870. Clarify.

12. Line 57 – authors mention three additional SST datasets and refer to Supplemental figure. It’s not clear which datasets were used for the original cluster analysis, nevertheless, which three additional datasets are used without reviewing the supplemental material. Clarify datasets.

13. Lines 62-65 – Confusing syntax. Consider revising for clarity on the two warming spots.

14. Line 140 – Confusing sentence and unclear what authors mean by AMO+ minus AMO-

15. Line 165 – The paleoclimate reconstruction dataset needs more attention and referenced.

16. Line 173 – Define what “EX” means

17. Line 196-198 – Wording is redundant from abstract and Line 211 – suggest changing wording slightly.

18. Line 349 – Capitalize L in Linear

19. Figure 2a – change “MEP” label to “EP”

References

Examples of other references to consider with respect to SST dataset limitations, more robust paleoclimate consideration, and previous AMO-ENSO connections. This is not exhaustive, but just an example and starting place in order to put results into context of the literature with respect to SST

dataset limitations when studying ENSO, additional paleoclimate studies, and ENSO-AMO studies.

1. Deser et al. 2009 – Sea Surface Temperature Variability: Patterns and Mechanisms

- Spatial limitation of data in extended SST products due to limited data availability in the tropical Pacific prior to 1960

2. Grothe et al. 2020 – Enhanced El Niño-Southern Oscillation Variability in Recent Decades

- Found similarly that the most recent decades are more intense but that the higher internal variability in the early 1900's was not significantly higher than preindustrial levels.

3. Wang et al., 2019 – Three-ocean interactions and climate variability: a review and perspective

- Review paper that demonstrates the warm AMO phase increases the occurrence of CP El Niños

Reviewer #2:

Remarks to the Author:

Review of NCOMMS manuscript by Gan et al.

The study explores whether the observed changes in the El Niño Southern Oscillation frequency since the 1980s can be attributed to natural variability or external forcing. The study uses observations, reanalysis, and paleo-proxies to compare an epoch when El Niño had similar frequency and temporal evolutions to those in the past 40 years. Based on this comparison, the study shows that the recent changes in El Niño events are primarily due to internal variability associated with the Atlantic Multidecadal Oscillation (AMO), but anthropogenic forcing has contributed to a less extent to the increase in extreme and Central Pacific El Niño events.

I find the study interesting and well-structured, and the results relevant to the scientific community. It helps put into context changes associated with internal variability versus external forcing on El Niño events, and it is important to consider for future ENSO projections. I have only one main point related to the classification of ENSO.

The study separates El Niño events into 4 categories based on a cluster analysis: Eastern Pacific (EP) El Niño, Successive El Niño, Extreme El Niño, Central Pacific (CP) El Niño. There is a wide spectrum of ENSO structures, and many other types of El Niño have been classified, such as Coastal El Niño (Garreaud 2018) and the Mixed type (Kug et al. 2009). Sub-classifying El Niño events may present a challenge for obtaining significant results of ENSO frequency changes as it reduces the sample size. Traditional analyses have separated El Niño into two broad types that are more widely used by the scientific community: EP and CP (e.g. Kao and Yu 2009; Capotondi et al. 2020). I wonder if it would be more beneficial to simplify El Niño categories to the more traditional 2 clusters (i.e. EP and CP)? For example, Cai et al. 2021 define EP El Niño as Extreme El Niño, based on their classification. This simpler classification would increase the sample size and potentially give more robust results. How do the results change if considering only two El Niño types? The authors have partially addressed this by testing the sensitivity of their classification analysis using other methods (e.g. Takahashi et al. 2011, described in Supplementary text on page 2). Do you find similar results?

References:

Kao, H.-Y., and J.-Y. Yu, 2009: Contrasting eastern-Pacific and central-Pacific types of ENSO. *Journal of Climate*, 22, 615–632.

Capotondi, A., A. T. Wittenberg, J.-S. Kug, K. Takahashi, and M. J. McPhaden, 2020: ENSO Diversity. *El Niño Southern Oscillation in a Changing Climate*, American Geophysical Union (AGU), 65–86, DOI: 10.1002/9781119548164.ch4

Takahashi, K., A. Montecinos, K. Goubanova, and B. Dewitte, 2011: ENSO regimes: Reinterpreting the canonical and Modoki El Niño. *Geophysical Research Letters*, 38, L10704, <https://doi.org/10.1029/2011GL047364>.

Garreaud, R. D. (2018). A plausible atmospheric trigger for the 2017 coastal El Niño, *International Journal of Climatology*, 38, e1296–31302. doi: 10.1002/joc.5426

Kug, J.-S., F.-F. Jin, & S.-I. An (2009). Two types of El Niño: Cold tongue El Niño and warm pool El Niño. *J. Climate*, 22, 1499–1515.

Cai, W., and Coauthors, 2021: Changing El Niño–Southern Oscillation in a warming climate. *Nat Rev Earth Environ*, 2, 628–644, <https://doi.org/10.1038/s43017-021-00199-z>.

Minor

It is worth discussing ENSO internal variability aspects as per Wittenberg (2009):

Wittenberg, A. T., 2009: Are historical records sufficient to constrain ENSO simulations? *Geophys. Res. Lett*, 36, L12702, <https://doi.org/10.1029/2009GL038710>.

Expand acronyms when first used: L.24 – SST, L.60 SSTA, L.113 WP, L.270 AA.

L.32-33, L.48: the authors refer to “anthropologic activities”. I am not an expert on English grammar/etymology, but I believe this should be “anthropogenic activities”, i.e. activities originated by humans.

L.63 and throughout the text: specify ‘boreal’ or ‘northern’ “spring” when referring to seasons. El Niño spans both hemisphere tropics.

L.129: “...North Atlantic resemble the AMO (Fig.3b).”

L.254, 262: “Niño3.4”

L.341: Is one member for each model used to calculate the multi-model mean?

L.343: “MME” instead of “MMM”?

L.349: “L”inear

Figure 1: The stippling indicates when the group mean is larger than one standard deviation from the group mean of each member. I am confused about what you refer to group mean and standard deviation. Panels a,b,d are means of 3 events, while panel e is the mean of 8 events. Is the standard deviation from 3 samples or the total El Niño samples?

Figure 2 caption: Specify the dashed line at the end of the timeseries in panel (a).

L.669-671: Specify what El Niño events are plotted in panels 2b-e. Do blue dots also represent Successive El Niño? Are Extreme El Niño plotted as stars? Double check the events; for example, 77 is outside the period specified in the figure caption (1875-1905 & 1980-2017) but still plotted as an orange diamond. Should it be blue? Or does this event refer to extreme EN 1877?

Fig.1a legend: Correct “MEP” to “EP”

Fig.3d: The increased frequency of Extreme and CP events during the positive phase of the AMO doesn’t seem immediately visible from Figure 3d. You could test if the number of extreme El Niño and CP El Niño is statistically different from the expected frequency during any other time in the reconstruction period, using for example a Monte Carlo test.

Supplementary material

L.4: Refer to figures in supplementary material as Supplementary Figure X instead of Figure X, as it can confuse it with the figures in the main manuscript.

Reviewer #3:

Remarks to the Author:

See attachment

Review of “Synchronized effects of greenhouse warming and internal variability have increased the frequency of extreme and central Pacific El Niño since 1980”

By analyzing observations and CMIP simulations, the authors show that the positive AMO may be responsible for the enhanced frequency of extreme and CP El Niño events, via enhancing the zonal sea surface temperature gradient in the CP and consequently strengthening zonal advective feedback. They first show that there are increased frequency of extreme and CP El Niño events over the periods of 1875-1905 and 1980-2017. They then show that both periods feature enhanced equatorial zonal SST gradient and are associated with a positive AMO. Next, they show that in 2/3 of CMIP models a positive AMO is associated with enhanced the equatorial zonal SST gradient which often leads to increased occurrence of extreme and CP El Niño. Finally, they analyze the relative contribution of external driver and internal variability.

I think the findings are novel and the analysis is overall convincing. I suggest minor revision with the following comments.

Better clarify the four types of El Niño:

The El Niño events are categorized into four types: 1) extreme, 2) EP, 3) CP, and 4) successive El Niño events.

Are these four types mutually exclusive? Can an EP or CP be extreme? or belong to one of the successive El Niño events? If not, how these four types are categorized actually?

According to supplementary information, “extreme” is defined when E-index is greater than 1.75 s.d., so does “extreme” only refer to “extreme EP”? and “EP” just refers to “moderate EP”?

I am confused. This information is important for understanding the whole study. Please clarify the four categories clearly in the main text.

Mechanisms underlying the relation between enhanced equatorial zonal SST gradient and more common extreme/CP El Niño events

It is not very clear to me through what processes the enhanced equatorial zonal gradient leads to more common extreme/CP El Niño events. While there is significant correlation and the authors mentioned enhance the zonal advective feedback process, the detail mechanisms are still a mystery. I wonder if the authors could say a little bit more on the potential mechanisms.

Are there any models that can reproduce this correlation between positive AMO and enhanced occurrences of CP and extreme El Niño. If so, this can be used to investigate the mechanism.

Abstract: more common extreme and Central Pacific (CP) El Niño events
This may be misunderstood as “extreme Central Pacific El Niño events”.

L143: larger than (). Some words are missing here.

L81-92: Is it really necessary to show the similar climate impacts? It might be more useful and direct to show the spatial pattern of the SST and precipitation anomaly in the tropics.

L155-164: Are there any previous studies on the AMO influence on the tropical SST that is consistent with the result here?

Figure 1: It would be helpful to add the results of EP El Niño, so the readers can clearly see the differences between EP and extreme/CP events.

Figure 2: It might be clearer to use figure legend to denote the meaning of orange diamond and black stars in b, just as the figure legend of a.

Discussion: I think it is worthwhile to talk about future changes in the equatorial zonal SST gradient and how that may change the El Niño. Models projected enhanced warming in the EP under global warming, this has significant impact on tropical precipitation pattern (e.g. Huang et al., 2013; Zhou et al., 2019) and would also influence the ENSO characteristics (e.g., Cai et al., 2021).

Cai, W., Santoso, A., Collins, M., Dewitte, B., Karamperidou, C., Kug, J.-S., Lengaigne, M., McPhaden, M. J., Stuecker, M. F., Taschetto, A. S., Timmermann, A., Wu, L., Yeh, S.-W., Wang, G., Ng, B., Jia, F., Yang, Y., Ying, J., Zheng, X.-T., ... Zhong, W. (2021). Changing El Niño–Southern Oscillation in a warming climate. *Nature Reviews Earth & Environment*, 2(9), 628–644.

Huang, P., Xie, S.-P., Hu, K., Huang, G., & Huang, R. (2013). Patterns of the seasonal response of tropical rainfall to global warming. *Nature Geoscience*, 6(5), 357–361.

Zhou, W., Xie, S.-P., & Yang, D. (2019). Enhanced equatorial warming causes deep-tropical contraction and subtropical monsoon shift. *Nature Climate Change*, 9(11), 834–839.

Response to reviewer

We wish to express our appreciation to the reviewers for all the insightful and constructive comments that helped us to improve the manuscript. We have addressed the comments from the three reviewers and revised the manuscript based on their suggestions. Please see our point-by-point response below. In the following, the reviewer's comments are written in blue, followed by our response in black.

Reply to Reviewer #1

General Comment:

Key Results

The authors are asking a very clear question – if a change in El Niño properties to more extreme and more Central Pacific (CP) events in the recent decades is due to natural variability or is it externally forced. They complete a cluster analysis of SST datasets to classify El Niño events and to compare similarities in events across time periods. Their main findings suggest there was another time period – 1875-1905 – with increased extreme and CP El Niño events, like the recent decades, but when anthropogenic CO₂ forcing was relatively lower. They then look CMIP6 climate models to investigate the internal variability influence on the decadal observations in changing El Niño properties. They find that the historical time period, as well as recent decades, coincides with a positive phase of the Atlantic Multidecadal Oscillation (AMO). Using a linear regression model, they estimate that internal variability (AMO) contributed to ~65% of the recent increasing extreme and CP El Niño events over the past four decades, with the anthropogenic forcing contributing to the remaining 35%.

Validity

The authors seem to have a reasonable interpretation of the data based on the results of the analyses performed. However, their results should be considered in more context alongside previous studies – particularly studies that have previously suggested AMO may modulate ENSO and paleoclimate studies that reconstruct ENSO pre-1950's (see a few suggested references below).

Significance

The results are highly significant to the field. There is growing evidence that El Niño events will not only become more extreme more frequently under future enhanced greenhouse gas warming, but that they are already observed to be more frequent in the recent decades. However, it's significant they highlight that natural variability still plays a role and can enhance the anthropogenic forcing. Additionally, the authors highlight another period where CP events were more common, similarly to today.

Data and methodology

The authors use three main SST datasets – the HadISST1, ERRST, and Kaplan. The reliability of these SST datasets for the equatorial Pacific reduces beyond 1950 despite the authors claim that they are reliable for the past 150 years (line 250). The authors state they create a “merged SST” dataset to reduce uncertainty. However, the paper gives little attention to uncertainties that may arise, affecting their pre-1950 data (see Deser et al., 2009). In the supplemental material, they complete a sensitivity test to illustrate that the classifications of El Niño events and the number of extreme El Niños of each SST dataset are identical to the merged SST product. However, this still falls

short of assessing the uncertainty related to limited data in the region of interest.

Response: We sincerely thank the reviewer for thoughtful comments and suggestions. In accordance with the reviewer's comments and suggestions, we have thoroughly revised the manuscript. In the revised manuscript, 1) we discussed the uncertainty in the SST products, 2) we added the information of previous ENSO-AMO and palaeoclimate studies in the Introduction section and added the discussion of the previous ENSO-AMO studies with our result, and 3) we clarified the data and method used in each section and improved the writing. We believe that the revised manuscript is significantly improved by addressing the reviewer's valuable comments.

Specific comments:

Suggested improvements

1. Give more attention to the uncertainty in the SST products and discuss the implications of uncertainties for the results.

Response: We acknowledge the uncertainty in the SST products especially prior to 1950 and fully agree with the reviewer that we should give more attention to the uncertainty in the SST products.

The uncertainties in the SST products can result from situ observations, analysis methods, and the choice of parameter values used to reconstruct or objectively analyze SSTs (Huang et al. 2017; Kennedy et al. 2011; Liu et al. 2015). Different SST products have different uncertainties (Huang et al. 2017). The observations prior to 1950 are relatively sparse. We acknowledge the limited data coverage in the tropical Pacific prior to 1960 (Deser et al. 2010). The three SST products (HadISST1, ERSSTV5 and Kaplan SST) that we used in this study based on in situ/satellite data have been produced employing different processing methods, such as interpolation method, smoothing method and so on (Huang et al. 2017; Kaplan et al. 1998; Rayner et al. 2003). The different processing method of the three SST products have produced different "perturbation" around the "truth". Therefore, we average the three datasets as a merged dataset to reduce the uncertainty, analogous to the reduction of error through multi-model averaging (L'Heureux et al. 2017).

To quantify the uncertainty, we perform the same analyses of El Niño classification based on the HadISST1, ERSSTv5, and Kaplan SST datasets (Supplementary Fig. 2a-c). In the three datasets, the numbers of extreme El Niño events are 3, 2, and 2 during the period 1870s to the 1900s, and 3, 3, and 3 during the period from the 1980s to the present. And that of CP El Niño events are 3, 3, and 3 in the first period and 7, 8, and 7 in the latter period. The difference among different datasets should arise from data uncertainty. Despite the slight difference, the increased extreme and CP event periods, 1870s to the 1900s and the 1980s to the present, are detectable from all the datasets, suggesting the result is not sensitive to the data uncertainty.

Moreover, we compare the climate impacts of El Niño events in the two periods. The composite of boreal winter air temperature anomaly patterns during extreme El Niño events in the period of 1875-1905 closely resembles those in the post-1980s, so do the CP El Niño events. The result also indicates that the detection of El Niño events from the SST dataset is reliable.

We have revised the paper according to the suggestion. The uncertainty among the three datasets and its impact on the result are discussed in the Lines 90-94. We have described the uncertainty of dataset in the section of Methods section (Lines 312-320), and we have deleted some improper sentences in the revised manuscript. We deleted the sentence “Since the observational SST records are reliable for approximately the past 150 years” (line 250 in the first revision) in the revised manuscript.

L90-94: “In the three datasets, the numbers of extreme El Niño events are 3, 2, and 2 during the period from the 1870s to the 1900s and 3, 3, and 3 during the period from the 1980s to the present, respectively. For CP El Niño, the number are 3, 3, and 3 during the period from 1870s to the 1900s and 7, 8, and 7 during the period from the 1980s to the present. The slight difference should arise from data uncertainty.”

L309-317: “Data uncertainties exist prior to 1950 due to the sparse coverage of instrumental observations across the equatorial Pacific⁵⁴. To reduce the uncertainty in the three SST datasets, we used a “merged monthly mean SST dataset (merged SST hereafter) following the pioneering work of ref.³⁴. The merged SST is defined as the arithmetic mean of the three monthly mean SST datasets. In addition to the “merged” SST dataset, we also used the HadISST1, ERSSTV5, and Kaplan SST datasets to examine the sensitivity of the results to different instrumental datasets (Supplementary Fig. 2). Although slight differences still exist, our key conclusion is not sensitive to the SST products, and it also provides some confidence to our result.”

References:

- Huang B, et al. Extended Reconstructed Sea Surface Temperature, Version 5 (ERSSTv5): Upgrades, Validations, and Intercomparisons. *J Climate* 30, 8179-8205 (2017).
- Kennedy JJ, Rayner NA, Smith RO, Parker DE, Saunby M. Reassessing biases and other uncertainties in sea surface temperature observations measured in situ since 1850: 1. Measurement and sampling uncertainties. *J Geophys Res Atmos* 116, (2011).
- Liu W, et al. Extended Reconstructed Sea Surface Temperature Version 4 (ERSST.v4): Part II. Parametric and Structural Uncertainty Estimations. *J Climate* 28, 931-951 (2015).
- Deser C, Alexander MA, Xie SP, Phillips AS. Sea surface temperature variability:

- patterns and mechanisms. *Ann Rev Mar Sci* 2, 115-143 (2010).
- Kaplan A, Cane MA, Kushnir Y, Clement AC, Blumenthal MB, Rajagopalan B. Analyses of global sea surface temperature 1856–1991. *Journal of Geophysical Research: Oceans* 103, 18567-18589 (1998).
- Rayner NA, et al. Global analyses of sea surface temperature, sea ice, and night marine air temperature since the late nineteenth century. *J Geophys Res Atmos* 108, (2003).
- L’Heureux ML, et al. Observing and Predicting the 2015/16 El Niño. *Bull Amer Meteor Soc* 98, 1363-1382 (2017).

2. Put results in context of previous studies. In the “References” section below, are a few suggested references that should be considered in the context of the results. Specifically, there seems to be little regard to previous ENSO-AMO and paleoclimate studies that have tried to address similar questions. For example, Line 35 states “... to the best of our knowledge, no study has done this work to date.” is not true. There is a plethora of paleoclimate studies that have tried to answer this question, and some that come to a different conclusion. Another example, Line 131 states “we suspect that the interdecadal variation in El Niño types was modulated by the AMO..” but does not reference any previous literature that already has investigated this.

Response: We are very thankful to the reviewer for directing us to the previous ENSO-AMO and palaeoclimate studies that have tried to address similar questions. We have added the information of previous ENSO-AMO and palaeoclimate studies (Grothe et al. 2020; Cobb et al. 2013; Gong et al. 2020; Zanchettin et al. 2016; Kang et al. 2014; Yu et al. 2015; Freund et al. 2015; Liu et al. 2017; Li et al. 2013) in the Introduction section (Lines 29-50) and have discussed our result in the context of previous ENSO-AMO studies in the section “Decadal variations in El Niño types modulated by the AMO” (Line 192-199) in the main text.

L29-50: “Several studies using climate models have projected that El Niño amplitude increases⁷ and CP El Niño variability increases¹¹ under greenhouse warming and have suspected that the observed El Niño behavior changes could be a consequence of anthropogenic warming. With the buildup of greenhouse gases, the eastern equatorial Pacific warmed faster than the surrounding regions, and the thermocline in the equatorial Pacific flattened, facilitating more occurrences of extreme El Niño and CP El Niño events^{7,11}. The enhanced response of atmospheric vapour to air temperature under greenhouse warming also increases the extreme El Niño¹⁸. Multiple palaeo-El Niño proxies also provide evidence for the impact of the climate change on the observed change in El Niño properties. These include enhanced El Niño variability during the late twentieth century relative to the preindustrial period^{12,19-21} and more frequent CP events in recent decades relative to the preindustrial era¹²⁻²². In addition to anthropogenic forcing, natural variability also plays a role in the change in El Niño properties²³⁻²⁴. Preindustrial model simulations show that El Niño characteristics exhibit strong interdecadal and

intercentennial modulation in the absence of external forcing²³. Palaeo-El Niño reconstructions also show a wide range of El Niño variance over the past 7000 year²¹ and find the El Niño variabilities in both the early 1900s and recent decades are relatively higher than preindustrial levels²⁵, highlighting the role of internal variability. Observations and preindustrial-control model simulations show that Atlantic Multidecadal Oscillation (AMO), a long-lived basin-wide warming or cooling in the Northern Atlantic that generally persists for 60–80 years, could modulate El Niño amplitude²⁶⁻²⁸ and El Niño type²⁹⁻³⁰.”

L191-198: “The spatial correlation field shows significant positive values over the western Pacific, suggesting that the equatorial zonal SST gradient is dominated by the western Pacific. Meanwhile, significant positive values are also found over the North Atlantic Ocean and the pattern resembles the AMO (Fig. 3a), suggesting that the zonal SST anomaly gradient is related to the AMO. This result is consistent with previous studies showing that the AMO could remotely affect the mean state of the Pacific⁴³⁻⁴⁶. Thus, the AMO seems to be a key candidate for leading to the multidecadal change in El Niño types.”

3. Improve overall clarity of writing. Below in the “Clarity and context” are more specific suggestions. The writing needs to be more specific with respect to datasets and methods used as it was not always clear what was being done or what datasets were being used. Sentences could be less wordy and more direct. For example, outline more specifics of the study design in the introduction. The results seem out of context until reading the end METHODS section.

Response: We thank the reviewer for the suggestion about clarity of writing. Following the suggestion of the reviewer, we tried to improve the clarity of writing from the following aspects:

First, we modified the sentences that did not clearly describe the datasets and methods used in the analysis. It includes the sentences reviewer mentioned in comments 9,12 and 15 of Reviewer #1 and some other sentences we found in this paper. The list is as follows:

L74-77: “We applied a cluster analysis to the evolution from the onset to the development of 38 El Niño events for the period 1871 to 2017 using a “merged” SST dataset from HadISST1³¹, Extended Reconstructed SST version 5 (ERSSTv5)³², and Kaplan Extended SST³³ (see Methods).”

L88-90: “The two increased extreme and CP event periods are also detectable from the HadISST1, ERSSTv5, and Kaplan SST datasets (Supplementary Fig. 2a-c).”

L97-98: “We compare the spatiotemporal evolution of extreme El Niño and CP El Niño events for the two periods using the “merged” SST dataset.”

L108-111: “Moreover, we conducted an ocean mixed-layer heat budget analysis of the mixed-layer ocean temperature averaged over the central-eastern Pacific (5°S-5°N, 180°-80°W) region during the El Niño development phase.”

L162-163: “Fig.2a shows the relationship between the Pacific zonal temperature gradient and El Niño types using the merged SST data.”

L182-183: “This result remains qualitatively unchanged if we use the HadISST1 ERSSTV5, and Kaplan SST datasets separately (Fig. 2c-e).”

L199-202: “We further analyzed the relationship between the AMO and El Niño diversity based on outputs from the fifth and sixth phases of the Coupled Model Intercomparison Project (CMIP5 and CMIP6). We use outputs from 20 CMIP5 and 23 CMIP6 pre-industrial control simulations over the last 300 years.”

L217-222: “We also use multicentury palaeoclimate reconstructions to examine the relationship between the AMO and the decadal modulation of the different types of El Niño events prior to the instrumental record. The palaeoclimate reconstructions include a 700-year (1300-2006) El Niño Niño3.4 index reconstruction²⁰, a 1200-year (800-2008) AMO index reconstruction⁴⁷, and a 400-year (1617-2008) record of CP El Niño events reconstructed from ENSO-sensitive proxy records¹².”

In Figure captions:

L848-851: “The merged SST data from 1871 to 2017 was used. The heat budget is calculated based on the merged Simple Ocean Data Assimilation (SODA) and Global Ocean Data Assimilation System (GODAS) data.”

L866: “The merged SST from 1871 to 2017 was used in a-b.”

L881: “The merged SST from 1871 to 2017 was used.”

L888: “20 CMIP5 and 23 CMIP6 pre-industrial control simulations were used.”

L892-894: “The CP El Niño events in the reconstruction are according to ref.12. The extreme El Niño events are defined by Niño-3.4 reconstruction index >1.2 standard deviations.”

L907: “The merged SST from 1871 to 2017 was used.”

Second, we checked the passive sentences in this paper and rephrased the unnecessary passive sentences with active voice. The list is as follows:

L16-17: We have rephrased “has been observed to change” to “has changed”.

L17-18: We have rephrased “have been frequently recorded” to “have occurred frequently”.

L89: We have rephrased “if the data are taken from three additional SST datasets” to “from the HadISST1, ERSSTv5, and Kaplan SST datasets”

L104: We have rephrased “Strong westerly anomalies are observed in the western Pacific” to “Strong westerly anomalies develop in the western Pacific”.

L262-263: We have rephrased “A linear regression model is used to estimate” to “We used a linear regression model to estimate”

L379: We have rephrased “the squared Euclidean distance was used to measure” to “we used the squared Euclidean distance to measure”

L411: We have rephrased “Pearson's chi-square test was used to test” to “We used Pearson's chi-square test to examine”

Third, we have added more information about the specifics of the study design in the introduction section (L51-L70) in the revised manuscript.

L51-L70: “It is still unclear whether the frequent occurrence of extreme El Niño events and CP El Niño events in the last 40 years is part of natural variability²³⁻²⁴ or a consequence of global warming^{1,7,11}. We can look back on the El Niño record in the past. If there exists a period that has frequent extreme El Niño and CP El Niño occurrences such as the past 40 years but is not affected much by anthropogenic activities, we can deduce that factors other than anthropogenic activities must play a role in the decadal transition of the El Niño type. In this study, we examine multiple long-term instrumental SST datasets of El Niño and apply cluster analysis to classify El Niño events. We show that there exists another extreme El Niño and CP El Niño epoch approximately 1900 with similar spatial and temporal evolutions, dynamic processes, and climate impacts as those that occurred in the last 40 years. The results suggest a role of internal variability. Moreover, we found that both the two periods with increased extreme El Niño and CP El Niño events coincide with the positive phase of the AMO. Then, we investigate the influence of internal variability associated with the AMO on El Niño multidecadal modulation using outputs from the fifth and sixth phases of the Coupled Model Intercomparison Project (CMIP5 and CMIP6) and palaeoclimate proxies. Furthermore, we quantify the contribution of anthropogenic forcing and internal variability to the recently observed El Niño diversity based on a statistical model. Our results highlight that both internal variabilities associated with the

AMO and anthropogenic forcing contribute to the changes in El Niño properties in recent decades.”

4. Improve overall organization of paper. The organization follows the authors’ logical thoughts, which is not necessarily the best way to present work (for examples – Line 132 – “To confirm this suspicion...” and Line 151 – “A natural question follows...”). The paper could be improved with describing what was done, what was found, and then discussing those results into context of other work, within each of the main sections. For example, Line 109 – “Here we show....” is presented before the authors state what they do. Consider walking the reader through the analyses and then end with the interpretation of how El Niño is tightly connected to zonal mean SST.

Response: Thank you for this constructive suggestion. Following the suggestion of the reviewer, we have reorganized the results section and reorganized Figure 3 and Figure 4 to improve the structure of this paper. We have rewritten the text in accordance with the principles of describing what was done, what was found, and discussing the results. We have modified Figure 1 as suggested by reviewers #3.

Particularly, the section of Decadal variations in El Niño types modulated by the AMO is rewritten as the reviewer kindly suggested:

L160-249: “What has caused the observed decadal transition of El Niño types? Previous studies suggest a relationship between changes in El Niño types and those in the Pacific zonal temperature gradient⁴⁰⁻⁴¹. Fig.2a shows the relationship between the Pacific zonal temperature gradient and El Niño types using the merged SST data. El Niño behavior is tightly connected with the variation in the zonal mean SST gradient (defined as the 31-year running mean of the difference between SSTA in the western Pacific [135°E–165°E] and SST in the eastern Pacific [175°W–145°W]) on a multidecadal time series: that is, the zonal gradient is stronger in the higher-frequency periods of extreme and CP El Niño events than in the higher-frequency period of EP El Niño events (Fig. 2a). Observation and model experiments suggest that the enhanced zonal SST gradient could provide a favorable condition for amplifying the zonal advective feedback, and the zonal advective feedback is a major dynamical feedback process, especially in the developing stage of El Niño initiated in the western Pacific^{10,40-41}. The enhanced zonal advective feedback over the CP is conducive to triggering the development of El Niño over the western Pacific region⁴². We noted that both extreme and CP El Niño events are associated with an initial warm anomaly in the western Pacific, while EP El Niño events are not (Fig. 1; Supplementary Fig. 3). We measure the initial development of El Niño over the western Pacific by calculating the observed western Pacific (130°E-170°W) SSTA during the El Niño onset phase (April through August). The western Pacific SSTA of El Niño is closely related to the mean-state zonal SST gradient ($r = 0.79$, $P < 0.001$, Fig. 2b), indicating that extreme El Niño and CP El Niño events rather than EP El Niño events tend to occur during periods of strengthened mean-state zonal SST gradients. This result remains

qualitatively unchanged if we use the HadISST1 ERSSTV5, and Kaplan SST datasets separately (Fig. 2c-e). Our results suggest that zonal SST gradient change is a controlling factor in determining the decadal transition of El Niño types.

Figure 3a shows the correlation of the SST field with the zonal SST anomaly gradient in the merged SST dataset. Here, the signal induced by external forcing, including the change in greenhouse gases (GHGs), natural forcing (NAT), and anthropogenic aerosols (AA), has been removed in the SST field and the zonal SST anomaly gradient (Fig. 3a, see Estimates of the forced and internal components in Method). The spatial correlation field shows significant positive values over the western Pacific, suggesting that the equatorial zonal SST gradient is dominated by the western Pacific. Meanwhile, significant positive values are also found over the North Atlantic Ocean and the pattern resembles the AMO (Fig. 3a), suggesting that the zonal SST anomaly gradient is related to the AMO. This result is consistent with previous studies showing that the AMO could remotely affect the mean state of the Pacific⁴³⁻⁴⁶. Thus, the AMO seems to be a key candidate for leading to the multidecadal change in El Niño types.

We further analyzed the relationship between the AMO and El Niño diversity based on outputs from the fifth and sixth phases of the Coupled Model Intercomparison Project (CMIP5 and CMIP6). We use outputs from 20 CMIP5 and 23 CMIP6 pre-industrial control simulations over the last 300 years. The CMIP5 and CMIP6 models could reproduce the observed AMO characteristics, with similar spatial patterns in the North Atlantic Ocean (Supplementary Fig. 6). Due to the model's bias in simulating the evolution of El Niño, we use the El Niño onset phase (April to August) averaged western Pacific SSTA $>0^{\circ}\text{C}$ to roughly distinguish the extreme/CP from EP events. A total of 30 of 43 CMIP5 and CMIP6 models (70%) simulated an increased western Pacific SST for the AMO-positive state minus the AMO-negative state (Fig.3b). The increased western Pacific SST enhances the zonal SST gradients, which is conducive to the development of El Niño in the Niño4 region^{10,40}. Models that generate a larger increase in the western Pacific SST/zonal SST gradient for the AMO-positive state minus the AMO-negative state tend to simulate a larger increase in the frequency of extreme/CP El Niño events for the AMO-positive state minus the AMO-negative state ($r = 0.66$, $P < 0.001$). A total of 29 of 43 CMIP5 and CMIP6 models (67.4%) produce an increased occurrence of CP and extreme El Niño for the AMO-positive state minus the AMO-negative state.

We also use multicentury palaeoclimate reconstructions to examine the relationship between the AMO and the decadal modulation of the different types of El Niño events prior to the instrumental record. The palaeoclimate reconstructions include a 700-year (1300-2006) El Niño Niño3.4 index reconstruction²⁰, a 1200-year (800-2008) AMO index reconstruction⁴⁷, and a 400-year (1617-2008) record of CP El Niño events reconstructed from ENSO-sensitive proxy records¹². To focus on the decadal modulation of different types of El Niño events, we count the occurrences of extreme El Niño events (defined by a Niño-3.4 index >1.2 s.d.) and CP Niño events (identified following the pioneering work

of ref.¹²) over a 21-year sliding period. The 21-year sliding frequencies of extreme El Niño events and CP Niño events increased in the positive phase of the AMO (Fig. 3c) and they are statistically significant above the 95% confidence level according to a bootstrap test (see Methods).

How does AMO impact El Niño diversity? We examined the process using the 5 CMIP6 and 3 CMIP5 models, in which the western Pacific SST responses to the AMO are comparable to the observations (Supplementary Fig. 7a). The simulated spatial pattern of the Pacific response to the AMO among these models exhibits a common feature: annual mean anticyclonic flows over the Northwest Pacific (NWP), significant warm SSTA over the western Pacific and the subtropical North Pacific (SNP), and strong northward flows from the tropics towards the SNP (Supplementary Fig. 7). We find that these features of the Pacific response to AMO derived from CMIP5 and CMIP6 are consistent with those derived from the observations (Supplementary Fig. 8) and a suite of Atlantic Pacemaker experiments⁴⁵. Such a spatial pattern of the Pacific response implies that the AMO could induce anomalous high pressure over SNP (Supplementary Fig. 8b), thereby causing SNP warming via wind–evaporation–SST feedback, and SNP warming could further develop warm SSTs in the western Pacific through SST–sea level pressure–cloud–longwave radiation positive feedback⁴⁵. The zonal SSTA gradient over the central Pacific increases as the western Pacific SSTA increases (Supplementary Fig. 9b), which in turn leads to enhanced zonal advection feedback in the development of El Niño (Supplementary Fig. 9c). Thus, a positive AMO corresponds to a warm SSTA in the western Pacific, a large zonal SSTA gradient over the central Pacific, and an enhanced CP and extreme El Niño, vice versa. These results imply that the AMO could modulate the frequency of the three types of El Niño events, with more frequent extreme and CP events in the positive phase of the AMO.”

Clarity and context

Overall, this paper could improve in clarity – in terms of both content and writing. It’s not always clear what the authors are referring to or what datasets they are using for the analyses. Additionally, most sentences are plagued with wordy passive tense and could be improved with a more active tense approach.

Response: We appreciate the reviewer’s suggestion about clarity and passive tense. Following the suggestion of the reviewer, we have improved the clarity and context from the following aspects:

We have checked the passive sentences in this paper and rephrased the unnecessary passive sentences with active voice. These rephrased sentences are listed in our response to comment 3 of Reviewer #1. We have modified the sentences that do not clearly describe the datasets and methods used in the analysis. It includes the sentences that reviewer mentioned in comment 9,12 and 15 of Reviewer #1 and some other sentences we found in this paper. Please see the detailed responses to comment 3

of Reviewer #1.

Below are specific suggestions to improve clarity:

5. Line 19 – disaster and disastrous are in the same since which is redundant; maybe change to “severe climatic disruptions...”

Response: Corrected. (Please see L19 in the revised manuscript).

6. Line 22 – “being located” is awkward; remove and just say, “which is characterized by the peak ocean warming in the central equatorial Pacific...”

Response: Corrected. (Please see L23 in the revised manuscript).

7. Line 24 – “being located” is awkward; remove “being” and say “..., SST anomalies located in the far eastern equatorial Pacific...”

Response: Corrected. (Please see L25 in the revised manuscript).

8. Line 35 – To the reader it is unclear what the authors are referring to here with “records”. Are they looking at paleo records? Or instrument SST records? Just in the equatorial Pacific? Or at other regions with teleconnections? What do they mean by “using multiple long-term instrumental SST datasets”? I think a better way to clarify that they are looking at SST products, the authors could rephrase to “we examine multiple long-term instrumental SST datasets of El Niño...”

Response: Thanks. We are looking at the instrument SST records. We have rephrased this sentence into “we examine multiple long-term instrumental SST datasets of El Niño... (Please see L57 in the revised manuscript)”.

9. Line 46 – Define what datasets the cluster analysis applied to.

Response: We have added the datasets that the cluster analysis applied to in the revised manuscript:

L74-77: “We applied a cluster analysis to the evolution from the onset to the development of 38 El Niño events for the period 1871 to 2017 using a “merged” SST dataset from HadISST1³¹, Extended Reconstructed SST version 5 (ERSSTv5)³², and Kaplan Extended SST³³ (see Methods)”

10. Line 48 –This statement is confusing as they list the cluster analysis years form 1871-2017 but say it’s the period “not affected much by anthropogenic activities”. Suggest rewording for clarity.

Response: We have deleted “not affected much by anthropogenic activities” and rephrased this statement as “We applied a cluster analysis to the evolution from the

onset to the development of 38 El Niño events for the period 1871 to 2017 using a “merged” SST dataset from HadISST1³¹, Extended Reconstructed SST version 5 (ERSSTv5)³², and Kaplan Extended SST³³ (see Methods).” (Please see L74-77 in the revised manuscript).

11. Line 51 – Confusing why the classifications were completed for 1901-2017, and not 1870. Clarify.

Response: We apologize for the insufficient information. In this study, we applied the cluster analysis and completed the classifications for 1871-2017. However, in the pioneering work of Wang et al., 2019, they completed the classifications for the period from 1901 to 2017. Therefore, we compared the classifications we obtained with those classifications shown in Wang et al., (2019) over the same period of 1901-2017.

To clarify this point, we have rephrased this sentence into “The four types of identified El Niño events are consistent with ref³⁴ during the common period (1901-2017). (Please see L81-82 in the revised manuscript).

12. Line 57 – authors mention three additional SST datasets and refer to Supplemental figure. It’s not clear which datasets were used for the original cluster analysis, nevertheless, which three additional datasets are used without reviewing the supplemental material. Clarify datasets.

Response: We have clarified datasets and rephrased this sentence into “The two increased extreme and CP event periods are also detectable from the HadISST1, ERSSTv5, and Kaplan SST datasets (Supplementary Fig. 2a-c).” (Please see L88-L90 in the revised manuscript).

13. Lines 62-65 – Confusing syntax. Consider revising for clarity on the two warming spots.

Response: Thanks. We have modified the relevant text (L97-L104).

L97-104: “We compare the spatiotemporal evolution of extreme El Niño and CP El Niño events for the two periods using the “merged” SST dataset. The evolution of the extreme El Niño events in the two periods is similar: the warm SST anomaly starts to develop in the western Pacific (WP) during the preceding boreal winter and then propagates eastward with a rapid basin-wide development in the boreal spring. Later, a warm anomaly occurs in the far eastern Pacific during boreal spring and then propagates westward. The maximum intensity occurs around 120°W in December (Fig. 1a-b; Supplementary Fig. 3a).”

14. Line 140 – Confusing sentence and unclear what authors mean by AMO+ minus AMO-

Response: Thanks, we have modified the relevant text in the revised manuscript:

L206-208: “A total of 30 of 43 CMIP5 and CMIP6 models (70%) simulated an increased western Pacific SST for the AMO-positive state minus the AMO-negative state (Fig.3b).”

L210-214: “Models that generate a larger increase in the western Pacific SST/zonal SST gradient for the AMO-positive state minus the AMO-negative state tend to simulate a larger increase in the frequency of extreme/CP El Niño events for the AMO-positive state minus the AMO-negative state ($r = 0.66$, $P < 0.001$).”

15. Line 165 – The paleoclimate reconstruction dataset needs more attention and referenced.

Response: To clarify this point, we have added a few sentences as “We also use multicentury palaeoclimate reconstructions to examine the relationship between the AMO and the decadal modulation of the different types of El Niño events prior to the instrumental record. The palaeoclimate reconstructions include a 700-year (1300-2006) El Niño Niño3.4 index reconstruction²⁰, a 1200-year (800-2008) AMO index reconstruction⁴⁷, and a 400-year (1617-2008) record of CP El Niño events reconstructed from ENSO-sensitive proxy records¹².” (Please see L217-222 in the revised manuscript).

16. Line 173 – Define what “EX” means

Response: We have added the definition of EX as “The externally forced (EX)” (Please see L253 in the revised manuscript”).

17. Line 196-198 – Wording is redundant from abstract and Line 211 – suggest changing wording slightly.

Response: We have rephrased this statement as “The linear regression model result shows that anthropogenic forcing-induced warming accounts for up to ~1 more extreme and ~2 more CP El Niño events from the 1980s to the present.” (Please see L276-278 in the revised manuscript). We have also rephrased the statement in discussion section as “However, anthropogenic forcing can explain ~1 more extreme and ~2 more CP events in the last 40 years.” (Please see L292-293 in the revised manuscript)

18. Line 349 – Capitalize L in Linear

Response: Corrected. (Please see L440 in the revised manuscript).

19. Figure 2a – change “MEP” label to “EP”

Response: Corrected.

References

Examples of other references to consider with respect to SST dataset limitations, more robust paleoclimate consideration, and previous AMO-ENSO connections. This is not exhaustive, but just an example and starting place in order to put results into context of the literature with respect to SST dataset limitations when studying ENSO, additional paleoclimate studies, and ENSO-AMO studies.

1. Deser et al. 2009 – Sea Surface Temperature Variability: Patterns and Mechanisms
 - Spatial limitation of data in extended SST products due to limited data availability in the tropical Pacific prior to 1960
2. Grothe et al. 2020 – Enhanced El Niño-Southern Oscillation Variability in Recent Decades
 - Found similarly that the most recent decades are more intense but that the higher internal variability in the early 1900’s was not significantly higher than preindustrial levels.
3. Wang et al., 2019 – Three-ocean interactions and climate variability: a review and perspective
 - Review paper that demonstrates the warm AMO phase increases the occurrence of CP El Niños

Response: We are very thankful to the reviewer for directing us to the previous studies on SST dataset limitations, more robust palaeoclimate consideration, and previous AMO-ENSO connections. These references and other related references were added in the revised manuscript.

Reply to Reviewer #2

General Comment:

The study explores whether the observed changes in the El Niño Southern Oscillation frequency since the 1980s can be attributed to natural variability or external forcing. The study uses observations, reanalysis, and paleo-proxies to compare an epoch when El Niño had similar frequency and temporal evolutions to those in the past 40 years. Based on this comparison, the study shows that the recent changes in El Niño events are primarily due to internal variability associated with the Atlantic Multidecadal Oscillation (AMO), but anthropogenic forcing has contributed to a less extent to the increase in extreme and Central Pacific El Niño events.

I find the study interesting and well-structured, and the results relevant to the scientific community. It helps put into context changes associated with internal variability versus external forcing on El Niño events, and it is important to consider for future ENSO projections. I have only one main point related to the classification of ENSO.

Response: We would like to express our appreciation to the reviewer for the constructive comments that helped us to improve the manuscript. In accordance with the reviewer's comments and suggestions, we have thoroughly revised the manuscript. In the revised manuscript, 1) we used the common Niño method to categorize El Niño and compared the results according to the Niño method with the results according to cluster method, 2) we discussed the ENSO internal variability, and 3) we added the Monte Carlo test to test whether the number of extreme El Niño/CP El Niño is statistically different in positive phase of AMO and negative phase of AMO. We also modified the Figure and caption as the reviewer suggested. We believe that the results from using the common classification method help to further convince readers of our conclusions.

Specific Comments:

1. The study separates El Niño events into 4 categories based on a cluster analysis: Eastern Pacific (EP) El Niño, Successive El Niño, Extreme El Niño, Central Pacific (CP) El Niño. There is a wide spectrum of ENSO structures, and many other types of El Niño have been classified, such as Coastal El Niño (Garreaud 2018) and the Mixed type (Kug et al. 2009). Sub-classifying El Niño events may present a challenge for obtaining significant results of ENSO frequency changes as it reduces the sample size. Traditional analyses have separated El Niño into two broad types that are more widely used by the scientific community: EP and CP (e.g. Kao and Yu 2009; Capotondi et al. 2020). I wonder if it would be more beneficial to simplify El Niño categories to the more traditional 2 clusters (i.e. EP and CP)? For example, Cai et al. 2021 define EP El Niño as Extreme El Niño, based on their classification. This simpler classification would increase the sample size and potentially give more robust results. How do the results change if considering only two El Niño types? The authors have

partially addressed this by testing the sensitivity of their classification analysis using other methods (e.g. Takahashi et al. 2011, described in Supplementary text on page 2). Do you find similar results?

References:

- Kao, H.-Y., and J.-Y. Yu, 2009: Contrasting eastern-Pacific and central-Pacific types of ENSO. *Journal of Climate*, 22, 615–632.
- Capotondi, A., A. T. Wittenberg, J.-S. Kug, K. Takahashi, and M. J. McPhaden, 2020: ENSO Diversity. *El Niño Southern Oscillation in a Changing Climate*, American Geophysical Union (AGU), 65–86, DOI: 10.1002/9781119548164.ch4
- Takahashi, K., A. Montecinos, K. Goubanova, and B. Dewitte, 2011: ENSO regimes: Reinterpreting the canonical and Modoki El Niño. *Geophysical Research Letters*, 38, L10704, <https://doi.org/10.1029/2011GL047364>.
- Garreaud, R. D. (2018). A plausible atmospheric trigger for the 2017 coastal El Niño, *International Journal of Climatology*, 38, e1296–31302. doi: 10.1002/joc.5426
- Kug, J.-S., F.-F. Jin, & S.-I. An (2009). Two types of El Niño: Cold tongue El Niño and warm pool El Niño. *J. Climate*, 22, 1499–1515.
- Cai, W., and Coauthors, 2021: Changing El Niño–Southern Oscillation in a warming climate. *Nat Rev Earth Environ*, 2, 628–644, <https://doi.org/10.1038/s43017-021-00199-z>.

Response: We thank the reviewer for the constructive suggestions. We fully agree with reviewer that sub-classifying El Niño events will reduce the sample sizes and we should examine how the results will change when we use the traditional 2 clusters. Compared to the traditional method, the cluster method we used in this study considers the temporal-spatial structure from the pre-onset to development processes rather than only the spatial structure at mature phase and distinguish the strong from moderate events.

To check the sensitivity of our result to method, follow the suggestion, we further used the traditional El Niño classification method (Kug et al 2009; Ham and Kug, 2012; Yeh et al. 2009), to categorize 38 El Niño events into EP and CP types. In this method, an El Niño event is classified as a CP (EP) type when the DJF-averaged value of the normalized Niño 4 index is greater (less) than the average value of the normalized Niño3 index. In such classification, CP El Niño occur frequently in 1875-1905 and 1980-present, consistent with our result (Fig R1a). The difference likely arises from that our cluster method not only consider SST spatial pattern in El Niño peak phase but also consider SST evolution in El Niño development (Feng et al. 2014).

As this traditional classification method cannot identify extreme El Niño event directly, we further divide the EP type El Niño events into moderate EP and extreme El Niño events by the criterion of DJF-averaged value of the normalized Niño3 index below or above a threshold value of 1.75 standard deviations. We found that extreme El Niño events occur more frequently in 1875-1905 and 1980-present (Fig.R1b). This result suggests that the two increased extreme and CP event periods is still detectable when the traditional classification method is used. We also did this analysis using the HadISST1, ERSSTv5, and Kaplan SST dataset, the two increased extreme and CP event

periods are also detectable (not shown).

We also calculate the contribution of internal variability and anthropogenic forcing to the change of El Niño types based on the traditional classification method. The internal variability contributed to 77 % of the increasingly extreme and CP El Niño events (~2.8 extreme and ~5.0 CP El Niño). Under anthropogenic forcing-induced warming, the frequency of occurrence of extreme and CP El Niño events has increased significantly, with ~1 more extreme and ~1 more CP El Niño events in the last 40 years. Thus, our key conclusions, that the recent changes in El Niño events were attributed to synchronized effects of greenhouse warming and internal variability, still holds when the traditional EP/CP classification is used.

Based on the comment of this reviewer, we have updated supplementary Figure 2, and reorganized supplementary Figure 10. We also added the supplementary Figure 11 and discussed the sensitivity to classification method in Lines 147-158 and Lines 278-281 in the main text and Lines 14-37 in the Supplementary information.

L147-158: “To examine the sensitivity of the extreme and CP El Niño event epoch of 1875- 1905 to the traditional classification method, we categorized 38 El Niño into two broad types (EP and CP) based on the Niño method^{10-11,39} (see Supplementary Text), and then we further divide the EP type El Niño events into moderate EP and extreme El Niño events by the criterion of DJF-averaged value of the normalized Niño3 index below or above a threshold value of 1.75 standard deviations (s.d.) (see Supplementary Fig. 2e and Supplementary Text). Based on the traditional classification method, we identified a total of 16 CP events and 8 extreme events, with 7 CP and 3 extreme events existing in the period from the 1980s to the present and 3 extreme and 3 CP events consecutively occurring in the period 1870s to the 1900s. Thus, the period of 1875-1905, like the recent decades, characterized by increased extreme El Niño and CP El Niño events are still detectable when employing the traditional classification method.”

L278-281: “Our result, that the recent changes in El Niño events were attributed to synchronized effects of greenhouse warming and internal variability, still holds when the traditional El Niño classification method is used (Supplementary Information gives a detailed analysis).”

Supplementary information, L14-37: “To test that the classification results are not sensitive to the classification method, we also used the traditional Niño method⁴⁻⁵ to categorize 38 El Niño events into CP and EP El Niño events. An El Niño event is classified as a CP (EP) type when the DJF-averaged value of the normalized Niño4 index is greater (less) than the average value of the normalized Niño3 index. Niño4 index equals the detrended SST anomaly averaged over 160°E–150°W, 5°S–5°N; Niño3 index equals the detrended SST anomaly averaged over 150°W–90°W, 5°S–5°N. However, this method cannot identify extreme El Niño event directly. Therefore, we further divide the EP type El Niño events into moderate EP and extreme El Niño events by the criterion of the DJF-averaged value of the

normalized Niño3 index below or above a threshold value of 1.75 standard deviations. We found that similar to the results based on the cluster analysis, there is another period (from the 1870s to 1900s) with increased CP and extreme El Niño events, like the recent decades. The difference likely arises from that our cluster method not only consider SST spatial pattern in El Niño peak phase but also consider SST evolution in El Niño development (Feng et al. 2014). In addition, the frequencies of different El Niño types based on the traditional classification method exhibit significant correlations with the internal multidecadal variability (Supplementary Fig. 10e-h), and these correlations are consistent with those based on the cluster classification method of this study (Supplementary Fig. 10). Moreover, we estimate that internal variability contributed to ~77% of the increasingly extreme and CP El Niño events, while anthropogenic forcing has made our globe experience ~1.1 more extreme and ~1.2 more CP events over the past four decades (Supplementary Fig. 11). Therefore, the conclusion still holds that the recent changes in El Niño events were attributed to synchronized effects of greenhouse warming and internal variability.”

Figure R1. Sensitivity of event classification to different classification method. **a** the occurrence of EP and CP types of El Niño events based on Niño method (Yeh et al. 2009; Kug et al 2009; Ham and Kug, 2012). The 38 El Niño events are shown in different color bars: CP (orange), and EP (blue). **b** the occurrence of extreme, EP and CP type of El Niño events. The CP events (orange) are the same as (a) but we separate the EP events in panel (a) into extreme events and the moderate events. The extreme events (red) is identified as when DJF-averaged of the normalized Niño3-index is larger than a threshold value of 1.75 standard deviations. The moderate EP events are shown

in solid blue circles. The time series of the 31-year running mean, annual-mean zonal SSTA gradient [western Pacific SSTW (135–165°E) minus central Pacific SSTC (165–145°W)] for the observations from 1871–2017 (°C, relative to the mean of 1901–2010, black line). The positive (Obs is greater than the 1901–2010 mean) and negative (Obs is less than the 1901–2010 mean) phases of the zonal equatorial SSTA gradient are represented by light red and light blue shading, respectively.

Supplementary Fig. 2 | Occurrence of different types of El Niño events. **a-c** sensitive to the SST datasets. **a** HadISST, **b** ERSSTv5, **c** Kaplan SST. The 38 El Niño events using different SST datasets are shown in different color bars: extreme (red), CP (orange), EP (blue), and Successive (gray). The black line is the time series of the 31-year running mean, annual-mean zonal equatorial SSTA anomalies (°C, relative to 1901–2010 mean) gradient (SSTA (5°S–5°N, 135°E–165°E) minus SSTA (5°S–5°N, 175°W–145°W) from 1871–2017. **d-e**, sensitive to classification method **d**, The extreme events (red) are based on E-index, the WCP (orange), EP (blue) and ECP (gray) are based on based on the pioneering work of ref.¹⁰. **e**, The extreme (red), EP (blue) and CP (orange)

events are based on Niño method.

Supplementary Fig. 10 | Simple linear regression model. **a** The relationship between the mean-state zonal SSTA gradient during the 5 different periods (1875-1905, 1906-1929, 1930-1949, 1950-1980, 1981-2017) marked in Fig. 2a and its corresponding ratio of the sum of extreme and CP (a) to total El Niño events. **b-d** similar to (a) but for the extreme (b), the CP (c), and the EP (d). **e-h** the same as a-d but the extreme, CP and EP events are identified based on Niño method. We used the HadISST, ERSSTV5, and Kaplan datasets shown in Supplementary Fig. 2a-c and the HadISST, ERSSTV5, and Kaplan merged SST datasets shown in Fig. 2a. The mean-state zonal SSTA gradient is calculated as the 31-year running mean, annual-mean zonal equatorial SST anomalies (°C, relative to 1901-2010 mean) gradient (SSTA (5°S–5°N, 135°E–165°E) minus SSTA (5°S–5°N, 175°W–145°W)) in five periods. The black line represents the fit line, and the blue band shows the 95% confidence interval.

Supplementary Fig.11 Same as Figure 4c but the extreme El Niño, CP El Niño and EP El Niño events are categorized according to the Niño method. Histograms for the estimated numbers of different types of El Niño events in 1981-2017 based on a linear regression model (see Methods and Supplementary Fig. 10e-f). The orange, blue, and red bars denote the EX+IV-, IV-, and EX-induced numbers of different types of El Niño events, respectively. Slant hatching denotes the AMO-induced numbers of different types of El Niño events.

References:

- Kug J-S, Jin F-F, An S-I. Two Types of El Niño Events: Cold Tongue El Niño and Warm Pool El Niño. *J Climate* **22**, 1499-1515 (2009).
- Ham Y-G, Kug J-S. How well do current climate models simulate two types of El Niño? *Climate Dyn* **39**, 383-398 (2012).
- Cai, W., and Coauthors, 2021: Changing El Niño–Southern Oscillation in a warming climate. *Nat Rev Earth Environ*, **2**, 628–644, <https://doi.org/10.1038/s43017-021-00199-z>.
- Feng JX, Wu ZH, Zou XL. Sea Surface Temperature Anomalies off Baja California: A Possible Precursor of ENSO. *J Atmos Sci* **2014**, **71**(5): 1529-1537.
- Yeh SW, Kug JS, Dewitte B, Kwon MH, Kirtman BP, Jin FF. El Niño in a changing climate. *Nature* **2009**, **461**(7263): 511-514.

Minor

2. It is worth discussing ENSO internal variability aspects as per Wittenberg (2009): Wittenberg, A. T., 2009: Are historical records sufficient to constrain ENSO simulations? *Geophys. Res. Lett*, **36**, L12702, <https://doi.org/10.1029/2009GL038710>.

Response: We are very thankful to the reviewer’s comments and suggestion. We have added a discussion in the Results section and added more information on the ENSO internal variability aspects in introduction section in the revised manuscript:

Result section: “It suggests that factors other than anthropogenic forcing must play a role in the decadal transition of El Niño types. This result is consistent with a model study showing that variations in El Niño behaviour can occur on multidecadal and intercentennial timescales even with fixed climate forcing

(Wittenberg et al 2009).” (Please see L142-146 in the revised manuscript).

Introduction section: “In addition to anthropogenic forcing, natural variability also plays a role in the change in El Niño properties (Wittenberg et al 2009; Newman et al 2011). Preindustrial model simulations show that ENSO characteristics exhibit strong interdecadal and intercentennial modulation in the absence of external forcing (Wittenberg et al 2009).” (Please see L40-44 in the revised manuscript).

Accordingly, we have also added the citation of the important paper (Wittenberg, A. T., 2009) recommended by the reviewer into the revised manuscript.

3. Expand acronyms when first used: L.24 – SST, L.60 SSTA, L.113 WP, L.270 AA.

Response: We have expanded the SST (L.26), SSTA (L.25), WP (L.100), AA (L.189) acronyms in the revised manuscript.

4. L.32-33, L.48: the authors refer to “anthropologic activities”. I am not an expert on English grammar/etymology, but I believe this should be “anthropogenic activities”, i.e. activities originated by humans.

Response: Corrected. (L.55-56, L.67-69)

5. L.63 and throughout the text: specify ‘boreal’ or ‘northern’ “spring” when referring to seasons. El Niño spans both hemisphere tropics.

Response: We have specified “boreal winter” and “boreal spring” in the revised manuscript.

6. L.129: “...North Atlantic resemble the AMO (Fig.3b).”

Response: We modify the sentence to “Meanwhile, significant positive values are also found over the North Atlantic Ocean and the pattern resembles the AMO (Fig. 3a), suggesting the zonal SST anomaly gradient is related to AMO.” (Please see L193-194 in the revised manuscript).

7. L.254, 262: “Niño3.4”

Response: Corrected.

8. L.341: Is one member for each model used to calculate the multi-model mean?

Response: Thanks for this comment. We apologize that our statement in the first round of revision are not clear enough. We used all ensemble runs for each model to calculate

the multi-model ensemble mean (MMM) as the externally forced signal. This method is based on the principle that the internal variations among the ensemble runs were usually uncorrelated. Thus, after averaging over a large number of ensemble simulations, the uncorrelated internal variations among the ensemble runs were largely smoothed out (Dai et al. 2015). To clarify this point, we have updated the relevant statement (Lines 425-428) in the revised manuscript.

L425-L428: “This method is based on the principle that the internal variations among the ensemble runs were usually uncorrelated. Thus, after averaging over a large number of ensemble simulations, the uncorrelated internal variations among the ensemble runs were largely smoothed out (Dai et al. 2015).”

References:

Dai A, Fyfe JC, Xie SP, Dai X. Decadal modulation of global surface temperature by internal climate variability. *Nat Climate Change* 2015, 5(6): 555-559.

9. L.343: “MME” instead of “MMM”?

Response: We have updated the acronyms for “multi-model ensemble mean” as “MMM” and have replaced the acronyms “MME” with the acronyms “MMM” in the revised manuscript. (Please see L422 in the revised manuscript).

10. L.349: “L”inear

Response: Corrected (L.440).

11. Figure 1: The stippling indicates when the group mean is larger than one standard deviation from the group mean of each member. I am confused about what you refer to group mean and standard deviation. Panels a,b,d are means of 3 events, while panel e is the mean of 8 events. Is the standard deviation from 3 samples or the total El Niño samples?

Response: Thank you for your comment. The corresponding group mean and standard deviation are calculated based on the different types of El Niño samples. For Fig.1a, the group mean is the mean of the three extreme El Niño samples and the standard deviation is from the three extreme El Niño samples. For Fig.1d, the group mean and the standard deviation is calculated from the 8 CP El Niño events.

To clarify this point, we have added some words in the revised manuscript: “The group mean and standard deviation are calculated based on the events used in each panel.” (Please see L836-837 in the revised manuscript).

12. Figure 2 caption: Specify the dashed line at the end of the timeseries in panel (a).

Response: Following the suggestion of the reviewer, we have added a description of the dashed line in the revised manuscript:

L863: “The dashed lines represent not full 31-year running.”

13. L.669-671: Specify what El Niño events are plotted in panels 2b-e. Do blue dots also represent Successive El Niño? Are Extreme El Niño plotted as stars? Double check the events; for example, 77 is outside the period specified in the figure caption (1875-1905 & 1980-2017) but still plotted as an orange diamond. Should it be blue? Or does this event refer to extreme EN 1877?

Response: Thanks for this comment. We apologize that the Figure 2 and the corresponding caption in the first round of revision are not clear enough and somewhat misleading. The orange diamond, black stars, and blue dots in Figure 2b-e represent the El Niño events occurred in the period of 1875-1905, the period of 1980-2017, and the period of 1906-1979, respectively.

As the period of 1875-1905 and the period of 1980-2017 are the main two periods which featured frequent occurrences of extreme El Niño events and CP El Niño events while the period of 1906-1979 are the period which featured frequent occurrences of EP El Niño events, we plotted the El Niño events occurred in the period of 1875-1905 as the orange diamond, the El Niño events occurred in the period of 1980-2017 as the black stars, and the El Niño events occurred in the period of 1906-1979 as the blue dots. The reason for using this is that we want to highlight that the extreme El Niño and CP El Niño events occurred in the period of 1875-1905 and occurred in period of 1980-2017 both follow the relationship that the observed western Pacific SSTA during the El Niño onset phase (April through August) indeed increases with the strengthened mean-state zonal SST gradient.

For the last question, the 77 refers to the extreme EN 1877; 84, 85, 88, 96, 99, 02, 04 refers to CP EN 1884, CP EN 1885, extreme EN 1888, CP EN 1896, EP EN 1899, extreme EN 1902, EP EN 1904, respectively.

Based on your comments, we have added a figure legend to denote the meaning of orange diamond and black stars in Figure 2 b-e and updated the Figure 2 caption and added the description of what El Niño events are plotted in panels 2b-e in the revised manuscript (Please also see L868-872 in the revised manuscript):

L868-872: “The black stars and orange diamonds in b-e represent the El Niño events that occurred from 1875 to 1905 (year 1877, 1884, 1885, 1896, 1899, 1902, 1904) and 1980 to 2017 (year 1982, 1986, 1991, 1994, 1994, 1997, 2002, 2004, 2006, 2009, 2014, 2015), respectively. Solid blue circles represent El Niño events that occurred from 1906 to 1979 (year 1911, 1913, 1918, 1923, 1925, 1930, 1957, 1963, 1965, 1968, 1972, 1976, 1977).”

14. Fig.1a legend: Correct “MEP” to “EP”

Response: Corrected.

15. Fig.3d: The increased frequency of Extreme and CP events during the positive phase of the AMO doesn't seem immediately visible from Figure 3d. You could test if the number of extreme El Niño and CP El Niño is statistically different from the expected frequency during any other time in the reconstruction period, using for example a Monte Carlo test.

Response: Thanks for the comment. Following the suggestion of the reviewer, we use the Monte Carlo bootstrapping to estimate the probability density function (PDF) of the number of the 21-year sliding frequency of extreme El Niño events and the number of the 21-year sliding frequency of CP El Niño during positive and negative phases of AMO. The resampling procedure was repeated 10 000 times. The 2.5 and 97.5% ranking from the probability distribution function are indicating the 95% confidence level. The result shows the distributions of both the number of extreme El Niño and the number of CP El Niño during positive phase of AMO are well-distinguished from that during the negative phase of AMO, with P values smaller than 0.05 (Figure R2 shown below). We have added the bootstrap test result in the revised manuscript:

L225-228: “The 21-year sliding frequencies of extreme El Niño events and CP Niño events increased in the positive phase of the AMO (Fig. 3c) and they are statistically significant above the 95% confidence level according to a bootstrap test (see Methods).”

L473-483: “We use a bootstrap test to examine whether the number of extreme El Niño/CP El Niño events is statistically different in the positive phase of AMO and negative phase of AMO. We use Monte Carlo bootstrapping to estimate the probability density function (PDF) of the number of the 21-year sliding frequencies of extreme El Niño events and the number of the 21-year sliding frequencies of CP El Niño during positive and negative phases of the AMO. The resampling procedure was repeated 10 000 times. The 2.5 and 97.5% rankings from the probability distribution function indicate the 95% confidence level. The number of extreme El Niño increased in positive phase of AMO compared to negative phase of AMO is statistically significant above the 95% confidence level if distributions of number of extreme El Niño during positive phase of AMO are well-distinguished from that during the negative phase of AMO.”

Figure R2. a The distribution of PDF of the number of CP El Niño estimates from 10 000 bootstrapped samples in the positive (orange) and negative (blue) phase of AMO index. Vertical dashed lines indicate the 5% and 95% confidence levels estimated using the percentile method. **b** the same as a but for extreme El Niño. The orange and blue dots indicate mean values of 10,000 inter-realizations for positive and negative phase of AMO index, respectively. The orange and blue horizontal lines denote the 1.0 s.d. of the 10,000 inter-realizations for positive and negative phase of AMO index, respectively.

16. Supplementary material

L.4: Refer to figures in supplementary material as Supplementary Figure X instead of Figure X, as it can confuse it with the figures in the main manuscript.

Response: Corrected (Please see L5 in Supplementary information).

Reply to Reviewer #3

General Comment:

By analyzing observations and CMIP simulations, the authors show that the positive AMO may be responsible for the enhanced frequency of extreme and CP El Niño events, via enhancing the zonal sea surface temperature gradient in the CP and consequently strengthening zonal advective feedback. They first show that there are increased frequencies of extreme and CP El Niño events over the periods of 1875-1905 and 1980-2017. They then show that both periods feature enhanced equatorial zonal SST gradients and are associated with a positive AMO. Next, they show that in 2/3 of CMIP models a positive AMO is associated with enhanced equatorial zonal SST gradients which often leads to increased occurrence of extreme and CP El Niño. Finally, they analyze the relative contribution of external driver and internal variability.

I think the findings are novel and the analysis is overall convincing. I suggest minor revision with the following comments.

Response: We would like to express our appreciation to the reviewer for the constructive comments that helped us to improve the manuscript. In accordance with the reviewer's comments and suggestions, we have thoroughly revised the manuscript. In the revised manuscript, 1) we clarified the four types of El Niño events, 2) we explained the mechanisms for the zonal SST gradient and the frequency of extreme/CP El Niño events, and 3) we added a discussion about the impact of future changes in the zonal gradients on ENSO characteristics. We also modified the Figure and caption as the reviewer suggested.

Specific comments:

1. Better clarify the four types of El Niño:

The El Niño events are categorized into four types: 1) extreme, 2) EP, 3) CP, and 4) successive El Niño events. Are these four types mutually exclusive? Can an EP or CP be extreme? or belong to one of the successive El Niño events? If not, how are these four types categorized actually? According to supplementary information, "extreme" is defined when E-index is greater than 1.75 s.d., so does "extreme" only refer to "extreme EP"? and "EP" just refer to "moderate EP"? I am confused. This information is important for understanding the whole study. Please clarify the four categories clearly in the main text.

Response: We thank the reviewer for this thoughtful suggestion. These four types are mutually exclusive. The EP and CP El Niño events cannot be the extreme El Niño or the successive El Niño events. The "EP" in this study just refers to "moderate EP" in more traditional 2 clusters (EP and CP). In the supplementary information, to test whether the classification based on the K-means cluster analysis is sensitive to the

classification method, we also used E- index to identify the extreme El Niño according to Cai et al. 2018 in this sensitivity analysis section.

Following the suggestion of this reviewer, we have added more information on the four categories to clarify this point in the revised manuscript:

L79-L81: “The four types of El Niño in this study are mutually exclusive. This categorization distinguishes extreme from moderate events, thus the “EP” in this study refers to “moderate EP”.”

2. Mechanisms underlying the relation between enhanced equatorial zonal SST gradient and more common extreme/CP El Niño events. It is not very clear to me through what processes the enhanced equatorial zonal gradient leads to more common extreme/CP El Niño events. While there is significant correlation and the authors mentioned enhance the zonal advective feedback process, the detail mechanisms are still a mystery. I wonder if the authors could say a little bit more on the potential mechanisms.

Response: We thank the reviewer for this constructive suggestion. The characteristics of El Niño are highly associated with the changes in the mean (or background) state of the tropical Pacific Ocean-atmosphere system (An and Wang 2000; Fedorov and Philander 2000; Fedorov et al. 2020; Ye and Hsieh 2006). The tropical mean state affects ENSO by altering major feedbacks that control the period and the growth rates of ENSO. Observational and model studies have suggested that zonal advection feedback is a major dynamical feedback process especially in the developing stage of El Niño initiated in the western Pacific (Xiang et al., 2013; Kug et al. 2009, 2010; Choi et al. 2010; Collins et al. 2010). Specifically, the zonal advective feedback is described as $-u' \partial \bar{T} / \partial x$ in a linear framework. The enhanced mean equatorial zonal SST gradient and the anomalous westerlies current favor the zonal advective feedback (Fig 2b-e). Thus, the enhanced zonal SST gradient could provide a favorable condition for amplifying the zonal advection feedback, which is conducive to El Niño being initiated in the western Pacific (Choi et al. 2010; Fang et al. 2018). Since most of CP and extreme El Niño events are characterized as initial developing in the western Pacific, the enhanced equatorial zonal gradient leads to more common extreme/CP El Niño events (Fig 2a).

Following the suggestion of this reviewer, we have added more information on the mechanisms for the zonal gradient effects on the extreme/CP El Niño in the Results section (Lines 169-174):

L169-L174: “Observation and model experiments suggest that the enhanced zonal SST gradient could provide a favorable condition for amplifying the zonal advective feedback, and the zonal advective feedback is a major dynamical feedback process, especially in the developing stage of El Niño initiated in the western Pacific^{10,40-41}. The enhanced zonal advective feedback over the CP is

conducive to triggering the development of El Niño over the western Pacific region^{40,42}.”

References:

- Ye Z, Hsieh WW. The influence of climate regime shift on ENSO. *Climate Dyn* 26, 823-833 (2006).
- An S-I, Wang B. Interdecadal Change of the Structure of the ENSO Mode and Its Impact on the ENSO Frequency*. *J Climate* 13, 2044-2055 (2000).
- Fedorov AV, Philander SG. Is El Niño changing? *Science* 288, 1997-2002 (2000).
- Fedorov AV, Hu S, Wittenberg AT, Levine AFZ, Deser C. ENSO Low - Frequency Modulation and Mean State Interactions. In: *El Niño Southern Oscillation in a Changing Climate* (2020).
- Choi J, An S-I, Kug J-S, Yeh S-W. The role of mean state on changes in El Niño’s flavor. *Climate Dyn* 37, 1205-1215 (2010).
- Kug J-S, Jin F-F, An S-I. Two Types of El Niño Events: Cold Tongue El Niño and Warm Pool El Niño. *J Climate* 22, 1499-1515 (2009).
- Kug J-S, Choi J, An S-I, Jin F-F, Wittenberg AT. Warm Pool and Cold Tongue El Niño Events as Simulated by the GFDL 2.1 Coupled GCM. *J Climate* 23, 1226-1239 (2010).
- Xiang B, Wang B, Li T. A new paradigm for the predominance of standing Central Pacific Warming after the late 1990s. *Climate Dyn* 41, 327-340 (2013).
- Collins M, et al. The impact of global warming on the tropical Pacific Ocean and El Niño. *Nature Geoscience* 3, 391-397 (2010).
- Fang X H, Mu M. A three-region conceptual model for central Pacific El Niño including zonal advective feedback[J]. *Journal of Climate*, 2018, 31(13): 4965-4979.

3. Are there any models that can reproduce this correlation between positive AMO and enhanced occurrences of CP and extreme El Niño. If so, this can be used investigate the mechanism.

Response: Following the suggestion, we examined the mechanism using the 5 CMIP6 and 3 CMIP5 models, in which the western Pacific SST responses to the AMO is comparable to the observations (Supplementary Fig. 7a). The simulated spatial pattern of the Pacific response to the AMO among these models exhibits a common feature: annual mean anticyclonic flows over the Northwest Pacific (NWP), significant warm SSTA over the western Pacific and the subtropical North Pacific (SNP), and strong northward flows from the tropics towards the SNP (Supplementary Fig. 7). We find that these features of the Pacific response to AMO derived from CMIP5 and CMIP6 are consistent with those derived from the observations (Supplementary Fig. 8) and a suite of Atlantic Pacemaker experiments (Sun et al. 2017). Such a spatial pattern of the Pacific response implies that the AMO could induce an anomalous high pressure over SNP (Supplementary Fig. 8b), thereby causing SNP warming via wind–evaporation–SST feedback, and SNP warming could further develop warm SSTs in the western Pacific through SST–sea level pressure–cloud–longwave radiation positive feedback

(Sun et al. 2017). The zonal SSTA gradient over the central Pacific enhance as the increase of in the western Pacific SSTA (Supplementary Fig. 9b), which in turn lead to enhanced zonal advection feedback in El Niño developing (Supplementary Fig. 9c). Thus, positive AMO correspond to warm SSTA in the western Pacific, large zonal SSTA gradient over the central Pacific, and enhanced CP and extreme El Niño, vice versa. These results imply that the AMO could modulate the frequency of the three types of El Niño events, with more frequent extreme and CP events in the positive phase of the AMO.

Based on the comment of this reviewer, we have updated supplementary Figure 7, and added the supplementary Figure 8-9 and discussed the mechanism in Lines 229-249 in the main text.

L229-249: “How does AMO impact El Niño diversity? We examined the process using the 5 CMIP6 and 3 CMIP5 models, in which the western Pacific SST responses to the AMO are comparable to the observations (Supplementary Fig. 7a). The simulated spatial pattern of the Pacific response to the AMO among these models exhibits a common feature: annual mean anticyclonic flows over the Northwest Pacific (NWP), significant warm SSTA over the western Pacific and the subtropical North Pacific (SNP), and strong northward flows from the tropics towards the SNP (Supplementary Fig. 7). We find that these features of the Pacific response to AMO derived from CMIP5 and CMIP6 are consistent with those derived from the observations (Supplementary Fig. 8) and a suite of Atlantic Pacemaker experiments⁴⁵. Such a spatial pattern of the Pacific response implies that the AMO could induce anomalous high pressure over SNP (Supplementary Fig. 8b), thereby causing SNP warming via wind–evaporation–SST feedback, and SNP warming could further develop warm SSTs in the western Pacific through SST–sea level pressure–cloud–longwave radiation positive feedback⁴⁵. The zonal SSTA gradient over the central Pacific increases as the western Pacific SSTA increases (Supplementary Fig. 9b), which in turn leads to enhanced zonal advection feedback in the development of El Niño (Supplementary Fig. 9c). Thus, a positive AMO corresponds to a warm SSTA in the western Pacific, a large zonal SSTA gradient over the central Pacific, and an enhanced CP and extreme El Niño, vice versa. These results imply that the AMO could modulate the frequency of the three types of El Niño events, with more frequent extreme and CP events in the positive phase of the AMO.”

Supplementary Fig. 8 | Pacific response to the AMO in Observation. **a** Regressions of Pacific SST (units: °C) on the normalized annual mean AMO index for 1871-2017 at decadal time scales. The green box represents subtropical North Pacific region (20°N–35°N, 170°E–155°W, SNP) and the black box represents Western Tropical Pacific region (5°S-25°N, 130°E-170°E, WTP). The merged SST is used. **b** the same as a but for ICOADS SLP (shading units: hPa) and for 1,000 hPa wind (vector, units: m s^{-1}). **c** the same as a but for cloud cover (units: %). **d**, the same as b but the SLP data derived from HadSLP dataset. All data are applied to a 21-year running mean filter. The linear trends are removed in all variables. For the wind, the National Centers for Environmental Prediction (NCEP) data were used.

Supplementary Fig. 9 | Mechanism for AMO forcing on El Niño diversity. **a** Inter-model relationship between the change in 21-year running mean subtropical North Pacific (20°S-35°N, 170°E-155°W, SNP) SSTA (x-axis, units: °C) and the change in 21-year running mean Western Tropical Pacific (5°S-25°N, 130°E-170°E, WTP) SSTA (y-axis, units: °C) for AMO-positive state minus AMO-negative state. **b** similar to **a** but for inter-model relationship between the change in 21-year running mean WP (5°S-5°N, 130°E-170°E) SSTA and 21-year running mean zonal SST gradient [western Pacific SST (5°S-5°N, 155°E-175°W) minus central Pacific SST (5°S-5°N, 115°W-145°W)]. **c** zonal advective feedback term of El Niño events during the development phase (April, May, June (AMJ)) averaged over the central-eastern Pacific (5°S-5°N, 180°-80°W) for AMO-positive state (AMO+, red bars) and AMO-negative state (AMO-, blue bars). The error bars in the multimodel mean represent the 95% confidence level determined by a bootstrap test. A total of 7 out of the 8 selected models (87.5%) simulate increased zonal advective feedback for El Niño events during AMO-positive state compared to AMO-negative state. Models that simulate a decrease are indicated by green circles. Models from CMIP5 are indicated in purple.

References:

Sun C, Kucharski F, Li J, Jin FF, Kang IS, Ding R. Western tropical Pacific multidecadal variability forced by the Atlantic multidecadal oscillation. *Nat Commun* **8**, 15998 (2017).

4. Abstract: more common extreme and Central Pacific (CP) El Niño events
This may be misunderstood as “extreme Central Pacific El Niño events”.

Response: Following the suggestion of the reviewer, we have modified the relevant text as “extreme El Niño and Central Pacific (CP) El Niño events” (Please see L3 in the revised manuscript).

5. L143: larger than (). Some words are missing here.

Response: Thanks, corrected and we delete this sentence due to other required revisions in the revised manuscript.

6. L81-92: Is it really necessary to show the similar climate impacts? It might be more useful and direct to show the spatial pattern of the SST and precipitation anomaly in the tropics.

Response: According to the suggestion, we have compared tropical SST anomaly between the two periods. The extremes El Niño events in the two periods show similar evolution characteristics (Figs 1. a and b), so do the CP El Niño events (Figs 1. d and e). The tropical SST spatial patterns of the two types of El Niño events are also comparable between the two periods (Figure R3). The result indicates that both the extreme El Niño and CP El Niño events are similar between the two periods.

As pointed out by the reviewer #1, SST dataset may exist uncertainty before 1950s. We need other datasets to valid the result. Most observational precipitation datasets mainly span the temporal domain after 1900 (Sun et al. 2018), which is not enough long for us to compare the similarities between the periods 1875-1905 and 1980-present. Since the observed land air temperature datasets covers the full temporal span 1875 to present and since different types of El Niño exert different impacts on the spatial pattern of land air temperature (Larkin, 2005), we still compare El Niño-related land air temperature pattern to valid our result.

Figure R3. Composites of sea surface temperature anomalies associated with El Niño diversity. a-b, Composite of developing year autumn (SON0) sea surface temperature anomalies (shading, °C) induced by extreme El Niño events for the periods of 1981-2017 (a) and 1875-1905 (b). c-d, same as a, b, but for the El Niño. The stippling denotes the regions where the signal (group mean) is larger than the noise (one standard deviation from the group mean of each member). The anomalies are linearly detrended. The HadISST1 data were used.

References:

Sun Q, Miao C, Duan Q, et al. A review of global precipitation data sets: Data sources, estimation, and intercomparisons. *Reviews of Geophysics*, 2018, 56(1): 79-107.
 Larkin NK. Global seasonal temperature and precipitation anomalies during El Niño autumn and winter. *Geophys Res Lett* 2005, 32(16).

7. L155-164: Are there any previous studies on the AMO influence on the tropical SST that is consistent with the result here?

Response: Thank you for your comment. The results are consistent with Sun et al. 2017, who used a suite of Atlantic Pacemaker experiments to investigate the AMO influence on the tropical western Pacific SST. Following the suggestion of the reviewer, we have modified the relevant text in the revision (Lines 235-238) about this point.

L235-238: “We find that these features of the Pacific response to AMO derived from CMIP5 and CMIP6 are consistent with those derived from the observations (Supplementary Fig. 8) and a suite of Atlantic Pacemaker experiments (Sun et al. 2017)”.

References:

Sun C, Kucharski F, Li J, Jin FF, Kang IS, Ding R. Western tropical Pacific multidecadal variability forced by the Atlantic multidecadal oscillation. *Nat Commun* 8, 15998 (2017).

8. Figure 1: It would be helpful to add the results of EP El Niño, so the readers can

clearly see the differences between EP and extreme/CP events.

Response: We thank the reviewer for the constructive comments. Following the suggestion of the reviewer, we have added the results of EP El Niño and updated the Figure 1 and the relevant statement in the revised manuscript (Lines 122-126):

L122-126: “The spatiotemporal evolution of EP El Niño events is different from those of extreme and CP El Niño events (Fig. 1g). The initial warm anomalies originate from the EP and then propagate westward. The thermocline feedback is dominant for EP events during the onset phase (Fig. 1h).”

Figure 1 | Composite evolution of the equatorial Pacific SST anomalies and heat budget analysis in extreme and CP El Niño for two periods: a-b The evolution of equatorial Pacific averaged (5°S - 5°N) SST anomalies (shading units: $^{\circ}\text{C}$) for extreme El Niño events during the period of (a) 1875-1905 and (b) 1981-2017. d-e Same as a-b but for CP El Niño events. g Same as a but for EP El Niño events during the period of 1875-2017. The stippling indicates the regions where the signal (group mean) is larger than the noise (one standard deviation from the group mean of each member). The group mean and standard deviation are calculated based on the events used in each panel. The anomalies are calculated referenced to the climatology of the full period and linearly detrended. c, f Comparison between the period of 1875-1905 and 1981-2017 in the ocean mixed-layer heat budget analysis of extreme El Niño (c) and CP El Niño (f) events during their respective onset phases (onset phases are defined as the month when the value of the Niño-3.4 index first exceeds 0.5°C and the two months after that) over the central-eastern Pacific (5°S - 5°N , 180° - 80°W). h Same as c but for EP El Niño

events during the period of 1875-2017. The 6 terms from left to right are $-u'\partial\bar{T}/\partial x$ (bar 1 denoted by ZA), $-\bar{u}\partial T'/\partial x$ (bar 2), $-u'\partial T'/\partial x$ (bar 3), $-w'\partial\bar{T}/\partial z$ (bar 4 denoted by EK), $-\bar{w}\partial T'/\partial z$ (bar 5 denoted by TH), and $-w'\partial T'/\partial z$ (bar 6). The units in the ordinates are $^{\circ}\text{C month}^{-1}$. The terms $-u'\partial\bar{T}/\partial x$, $-\bar{w}\partial T'/\partial z$, and $-w'\partial\bar{T}/\partial z$ denote the zonal advective feedback, thermocline feedback and upwelling feedback, respectively. The merged HadISST1, ERSST5 and Kaplan SST data from 1871 to 2017 was used. The heat budget is calculated based on the merged Simple Ocean Data Assimilation (SODA) and Global Ocean Data Assimilation System (GODAS) data.

9. Figure 2: It might be clearer to use figure legend to denote the meaning of orange diamond and black stars in b, just as the figure legend of a.

Response: Thanks. Following the suggestion of the reviewer, we have added the figure legend in Figure 2 and updated the Figure 2 in the revised manuscript.

Figure 2| The changing El Niño types from 1871 to 2017 and their relationship with the mean state zonal SSTA gradient. **a** The occurrence of different types of El Niño events. The 38 El Niño events are shown in different color bars: extreme (red), CP (orange), EP (blue), and Successive (gray). The time series of the 31-year running mean, annual-mean zonal SSTA gradient [western Pacific SST (135–165°E) minus central Pacific SST (165–145°W)] for the observations from 1871–2017 ($^{\circ}\text{C}$, relative to the mean of 1901-2010, black line). The positive (Obs is greater than the 1901-2010 mean) and negative (Obs is less than the 1901-2010 mean) phases of the zonal equatorial SSTA gradient are represented by light red and light blue shading, respectively. The dashed lines represent not full 31-year running. **b** The relationship between the mean-state zonal SSTA gradient and the western Pacific (WP) (130°E–170°W) SSTA in merged SST during the El Niño onset phase from April (0) to August (0). The merged SST from 1871 to 2017 was used in a-b. **c-e** Similar to **b** but for

HadISST (c), ERSSTV5 (d), and Kaplan (e). The black stars and orange diamonds in b-e represent the El Niño events that occurred from 1875 to 1905 (year 1877, 1884, 1885, 1896, 1899, 1902, 1904) and 1980 to 2017 (year 1982, 1986, 1991, 1994, 1994, 1997, 2002, 2004, 2006, 2009, 2014, 2015), respectively. Solid blue circles represent El Niño events that occurred from 1906 to 1979 (year 1911, 1913, 1918, 1923, 1925, 1930, 1957, 1963, 1965, 1968, 1972, 1976, 1977). The correlation (R) and the P value of linear regression (black solid line) are also shown. The mean state is defined by the 31-year running mean.

10. Discussion: I think it is worthwhile to talk about future changes in the equatorial zonal SST gradient and how that may change the El Niño. Models projected enhanced warming in the EP under global warming, this has significant impact on tropical precipitation pattern (e.g. Huang et al., 2013; Zhou et al., 2019) and would also influence the ENSO characteristics (e.g.,Cai et al.,2021).

Cai, W., Santoso, A., Collins, M., Dewitte, B., Karamperidou, C., Kug, J.-S., Lengaigne, M., McPhaden, M. J., Stuecker, M. F., Taschetto, A. S., Timmermann, A., Wu, L., Yeh, S.-W., Wang, G., Ng, B., Jia, F., Yang, Y., Ying, J., Zheng, X.-T., ... Zhong, W. (2021). Changing El Niño–Southern Oscillation in a warming climate. *Nature Reviews Earth & Environment*, 2(9), 628–644.

Huang, P., Xie, S.-P., Hu, K., Huang, G., & Huang, R. (2013). Patterns of the seasonal response of tropical rainfall to global warming. *Nature Geoscience*, 6(5), 357–361.

Zhou, W., Xie, S.-P., & Yang, D. (2019). Enhanced equatorial warming causes deep-tropical contraction and subtropical monsoon shift. *Nature Climate Change*, 9(11), 834–839.

Response: We are very thankful to the reviewer’s comments and suggestion. Our result show that extreme El Niño and CP El Niño events tend to occur in the period of an enhanced equatorial west-minus-east SST gradient. Thus, the projection of El Niño types should depend on the projection of equatorial west-minus-east SST gradient in the Pacific. Most climate models projected enhanced warming in the Eastern Pacific under global warming, which may imply that an increase in the frequency of EP El Niño events and a decrease in the frequency of extreme El Niño and CP El Niño events. However, the projection of mean state of equatorial Pacific still exist controversy, which hinder us to make a certain deduction here. Following the suggestion of the reviewer, we have added a discussion about the future changes in the “conclusion and discussion” section (Lines 297-303) in the revised manuscript:

L297-303: “Future change in El Niño is a crucial issue. Climate models projected a tropical Pacific mean state change under global warming, which could not only exert a significant impact on tropical precipitation patterns (Huang et al., 2013;

Zhou et al., 2019), but also influence the ENSO characteristics (Cai et al., 2021; Callahan et al., 2021). As extreme El Niño and CP El Niño events tend to occur in the period of an enhanced equatorial west-minus-east SST gradient, the projection of El Niño types should partly depend on the projection of equatorial west-minus-east SST gradient in the Pacific.”

References:

- Cai, W., Santoso, A., Collins, M., Dewitte, B., Karamperidou, C., Kug, J.-S., Lengaigne, M., McPhaden, M. J., Stuecker, M. F., Taschetto, A. S., Timmermann, A., Wu, L., Yeh, S.-W., Wang, G., Ng, B., Jia, F., Yang, Y., Ying, J., Zheng, X.-T., ... Zhong, W. (2021). Changing El Niño–Southern Oscillation in a warming climate. *Nature Reviews Earth & Environment*, 2(9), 628–644.
- Huang, P., Xie, S.-P., Hu, K., Huang, G., & Huang, R. (2013). Patterns of the seasonal response of tropical rainfall to global warming. *Nature Geoscience*, 6(5), 357–361.
- Zhou, W., Xie, S.-P., & Yang, D. (2019). Enhanced equatorial warming causes deep-tropical contraction and subtropical monsoon shift. *Nature Climate Change*, 9(11), 834–839.
- Callahan CW, Chen C, Rugenstein M, Bloch-Johnson J, Yang ST, Moyer EJ. Robust decrease in El Niño/Southern Oscillation amplitude under long-term warming. *Nat Climate Change* 11, 752–757 (2021).

Reviewers' Comments:

Reviewer #1:

Remarks to the Author:

I feel the authors successfully addressed my comments in my first review and I am satisfied with their changes. The science, organization, and writing is much improved and more clear.

Reviewer #2:

Remarks to the Author:

The authors addressed all my comments satisfactorily. I have no further comments. I believe this version of the manuscript has improved and I am happy for it to be accepted for publication.

Reviewer #3:

Remarks to the Author:

The authors have satisfactorily addressed all of my comments. I thus recommend acceptance.

Response to reviewer

We wish to express our appreciation to the reviewers for all the insightful and constructive comments that helped us to improve the manuscript. We have addressed the minor comments from the three reviewers. Please see our point-by-point response below. In the following, the reviewer's comments are written in blue, followed by our response in black.

Reply to Reviewer #1

General Comment:

I feel the authors successfully addressed my comments in my first review and I am satisfied with their changes. The science, organization, and writing is much improved and more clear.

Response: We would like to express our appreciation to the reviewer for the approval for our revision.

Reply to Reviewer #2

General Comment:

The authors addressed all my comments satisfactorily. I have no further comments. I believe this version of the manuscript has improved and I am happy for it to be accepted for publication.

Response: Thank you for your approval for our revision.

Reply to Reviewer #3

General Comment:

The authors have satisfactorily addressed all of my comments. I thus recommend acceptance.

Response: We sincerely thank the reviewer for the approval for our revision.